



# Quantification of $CH_4$ coal mining emissions in Upper Silesia by passive airborne remote sensing observations with the MAMAP instrument during CoMet

Sven Krautwurst[1], Konstantin Gerilowski[1], Jakob Borchardt[1], Norman Wildmann[2], Michal Galkowski[3,6], Justyna Swolkien[4], Julia Marshall[3], Alina Fiehn[2], Anke Roiger[2], Thomas Ruhtz[5], Christoph Gerbig[3], Jaroslaw Necki[6], John P. Burrows[1], Andreas Fix[2], and Heinrich Bovensmann[1]

[1]Institute of Environmental Physics (IUP), University of Bremen, Bremen, Germany
[2]Deutsches Zentrum für Luft- und Raumfahrt (DLR), Institut für Physik der Atmosphäre, Oberpfaffenhofen, Germany
[3]Department Biogeochemical Signals, Max Planck Institute for Biogeochemistry, Jena, Germany
[4]Faculty of Mining and Geoengineering, AGH University of Science and Technology, Krakow, Poland
[5]Institute for Space Sciences, Free University of Berlin, Berlin, Germany
[6]Faculty of Physics and Applied Computer Science, AGH University of Science and Technology, Krakow, Poland

**Correspondence:** Sven Krautwurst (krautwurst@iup.physik.uni-bremen.de)

**Abstract.**

Methane ($CH_4$) is the second most important anthropogenic greenhouse gas, whose atmospheric concentration is modulated by human-induced activities, and it has a larger global warming potential than carbon dioxide ($CO_2$). Because of its short atmospheric lifetime relative to that of $CO_2$, the reduction of the atmospheric abundance of $CH_4$ is an attractive target for short

term climate mitigation strategies. However, reducing the atmospheric $CH_4$ concentration requires a reduction of its emissions and, therefore, knowledge of its sources is essential.

For this reason, the $CO_2$ and Methane (CoMet) campaign in early summer of 2018 was initiated with the primary goal of assessing emissions of one of the largest $CH_4$ emission hot spots in Europe, the Upper Silesian Coal Basin (USCB) in southern Poland, using top-down approaches and inventory data. In this campaign, a variety of instruments (both in situ and remote

sensing) and platforms (e.g., ground-based and airborne) were deployed, which were supplemented by modeling activities supporting the flight planning and the interpretation of the observations. Consequently, $CH_4$ emissions originating from ~54 coal mine ventilation shafts distributed over an area of around $60 \times 40\,\mathrm{km^2}$ could be investigated on different scales, ranging from single shafts over smaller clusters up to the entire basin.

In this study, we will focus on $CH_4$ column anomalies retrieved from spectral radiance observations, which were acquired

by the 1D nadir-looking passive remote sensing Methane Airborne MAPper (MAMAP) instrument, using the Weighting Function Modified Differential Optical Absorption Spectroscopy (WFM-DOAS) method. The column anomalies are combined with wind lidar measurements and inverted to cross-sectional fluxes for different flight tracks making use of a mass balance approach. These fluxes are subsequently used to assess the reported emissions of small clusters of ventilation shafts.

The MAMAP $CH_4$ column observations allow for accurate assignment of observed fluxes to small clusters of ventilation

shafts. $CH_4$ fluxes are estimated for 4 clusters comprising 23 ventilation shafts in total, which are responsible for about 40 %





of the total $CH_4$ emissions from mining in the target area. The observations used were made during multiple overflights on different days between 28 May and 7 June 2018. The final averaged $CH_4$ fluxes for the single clusters (or sub-clusters) range from about 1 to $9\,t\,CH_4\,hr^{-1}$ at the time of the campaign. The range of fluxes observed at one cluster during different overflights can vary by as much as 50 % of the respective averaged value. Associated errors (1-$\sigma$) are usually between 15 % and

59 % of the averaged flux, mainly depending on the prevailing wind conditions, the number of flight tracks, and the magnitude of the flux itself. Comparison to known hourly emissions, where available, shows good agreement with the computed fluxes within the uncertainties. In the case that only annually reported emissions are available for comparison with the observations, caution is required due to potential fluctuations of the emissions during one year or even within hours. To measure emissions even more precisely and to further unravel them for allocation to individual shafts in a complex source region as encountered

in the USCB, imaging remote sensing instruments are recommended.

## 1   Introduction

The release of greenhouse gases from anthropogenic activity significantly influences the atmospheric surface temperature and the Earth's climate (Stocker et al., 2013). Consequently, there is a well recognized need to reduce these emissions (Fesenfeld

et al., 2018; UNFCCC, 2015, 1998). The largest impact on the surface temperature results from the increase in carbon dioxide ($CO_2$), which exerts a radiative forcing (RF) of ~$1.8\,W\,m^{-2}$ (Etminan et al., 2016). The second most important man made increase in radiative forcing results from the increase in methane ($CH_4$) with ~$0.6\,W\,m^{-2}$ (Etminan et al., 2016). However, on a per mass basis, $CH_4$ is 34 times more efficient in trapping heat in the Earth's atmosphere over 100 years than $CO_2$ (Myhre et al., 2013, including climate-carbon feedbacks). Moving to shorter time scales (e.g., 20 years), the effectiveness (or the global

warming potential, GWP) of $CH_4$ rises to 86 times that of $CO_2$ (Myhre et al., 2013, including climate-carbon feedbacks). The relatively high GWP of $CH_4$ in combination with a relatively short atmospheric lifetime of around 9 years (Prather et al., 2012) makes $CH_4$ an attractive target for short term emission and, thus, climate mitigation strategies (Saunois et al., 2016; Shindell et al., 2012).

To reduce methane emissions, their emission strengths and also locations need to be known. However, current knowledge

is inadequate as evidenced by the discussion about the origin of increasing atmospheric $CH_4$ concentrations observed since 2007 (Dlugokencky et al., 2011). Depending on the applied methodology (e.g., measuring ethane-to-methane ratio or isotopic analysis), authors either conclude that $CH_4$ emissions from fossil fuels (Franco et al., 2016; Hausmann et al., 2016; Helmig et al., 2016; Turner et al., 2016) or from wetlands and agriculture (Nisbet et al., 2016; Schaefer et al., 2016; Schwietzke et al., 2016) have increased or that the increase in atmospheric $CH_4$ is even related to a decline in atmospheric OH, which removes

the $CH_4$ (Rigby et al., 2017; Turner et al., 2017). Interestingly, even though Schwietzke et al. (2016) concluded that the increase is mostly related to wetlands and agriculture, they further stated that global emissions from the fossil fuel industry could be





~40 % higher than previously expected by Saunois et al. (2016). A study by Petrenko et al. (2017) supports this hypothesis and finds indications that even this revised number might be too low by at least 25 %. A recent study from Jackson et al. (2020) also concluded that the global increase in atmospheric $CH_4$ has been mostly driven by anthropogenic emissions and natural

$CH_4$ emissions remained almost unaltered between the period 2000–2006 and 2017. However, not only globally, but also on smaller scales our knowledge and characterization of fossil fuel $CH_4$ emissions is inadequate (e.g., Buchwitz et al., 2017; Maasakkers et al., 2016; Alexe et al., 2015; Turner et al., 2015).

A large source of anthropogenically emitted $CH_4$ originates from coal mining. It globally accounts for around one-tenth of the anthropogenic $CH_4$ emissions of about 350 $Mt\,CH_4\,yr^{-1}$ (Saunois et al., 2016, 2020). China, the largest emitter of $CH_4$

from coal mining, is responsible for ~50 % of the global total (EPA, 2012). The share of the European Union is around 4 %, with the largest contribution originating from Poland. This country is also home to the largest contemporary hard coal mining area in Europe, located in the Upper Silesian Coal Basin (USCB), occupying around 7400 $km^2$ (Gzyl et al., 2017) in total, and extending into the Czech Republic (compare to Fig. 1, area in Poland is around 5400 $km^2$).

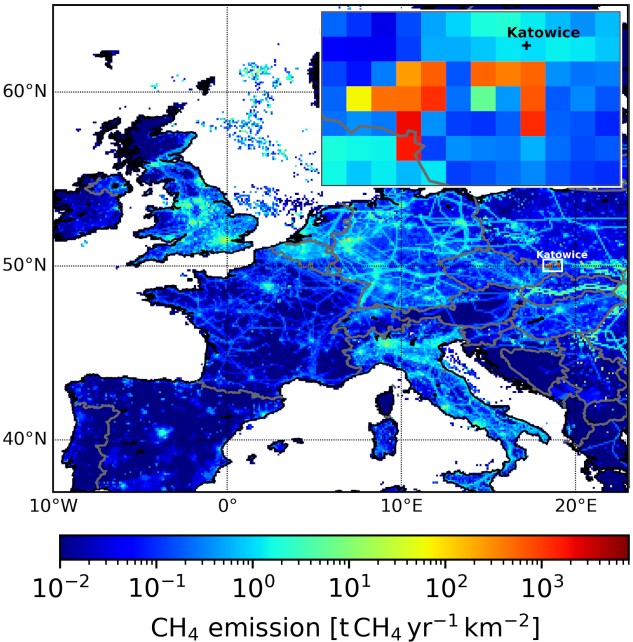

**Figure 1.** European $CH_4$ emissions from fossil fuels in 2016. The Upper Silesian Coal Basin (USCB) is located at around 19° E and 50° N framed by the white rectangle. A magnification is shown in the inset. Emission map is based on data from Scarpelli et al. (2020).



According to the latest bottom-up inventories (i.e. emissions calculated from emission factors and activity data), the EDGAR
v4.3.2[1] inventory for 2012 (Janssens-Maenhout et al., 2019) and v5.0[2] for 2015 (Crippa et al., 2020), and an inventory specially
designed for fossil fuel emissions from Scarpelli et al. (2020) for 2016, annual fossil fuel $CH_4$ emissions range from about 550
to $820\,\mathrm{kt}\,CH_4\,\mathrm{yr}^{-1}$ in that region. The largest contribution is attributed to coal mining activities, depending on the inventory
between 87 % (Crippa et al., 2020) and 99 % (Scarpelli et al., 2020).

To investigate this European $CH_4$ emission hot spot, the Carbon dioxide and Methane (CoMet) campaign was performed
in May and June 2018. One of its main goals was the estimation of $CH_4$ emissions from coal mining by using a top-down
approach and assess the available inventory data. Additionally, the synergistic value of having both airborne passive and active
remote sensing instruments was intended to be investigated. In this study we investigate the emissions from observations made
by the airborne passive remote sensing instrument MAMAP (Methane Airborne MAPper, Gerilowski et al., 2011) and wind
lidar observations for different groups of ventilation shafts. In terms of spatial scales, the resulting fluxes can be classified
between the already published analyses from this campaign, and characterize $CH_4$ emissions from the basin further. Nickl
et al. (2020) performed model simulations and Fiehn et al. (2020) computed fluxes from airborne in situ observations for the
entire basin, whereas Luther et al. (2019) estimated emissions from several shafts by means of mobile on ground FTS (Fourier
Transform Spectrometer) observations. Further studies including the synergistic use of instruments and models are planned as
part of the special issue "CoMet: a mission to improve our understanding and to better quantify the carbon dioxide and methane
cycles".

This manuscript is organized as follows. Section 2 introduces the applied methods. This comprises a more comprehensive
description of the CoMet campaign including the instrumentation (Sect. 2.1), how the passive remote sensing (Sect. 2.2.1) and
the wind lidar (Sect. 2.2.2) observations are processed and used to compute cross-sectional fluxes (Sect. 2.2.3). Those fluxes
are then assigned to different mining clusters in Sect. 2.3 and Sect. 2.4 describes the inventory used for comparison. In section
3 the results are presented, including the general wind situation on the different flight days (Sect. 3.1) and the computed fluxes
for the different mining clusters (Sect. 3.2). Section 4 provides a more detailed comparison and discussion on the computed
fluxes and reported $CH_4$ emissions. Finally, the results are discussed in Sect. 5 and conclusions are drawn in Sect. 6.

## 2 Methods and data

The activities and set-up of the CoMet campaign are described below. This comprises an overview of the target area, the
investigated coal mine ventilation shaft complex, and the deployed instrumentation. A more detailed description of the airborne
passive remote sensing instrument MAMAP and the wind lidar stations are provided. The retrieval algorithm(s) and the method
used to determine the cross-sectional fluxes, including expected errors are presented. Finally, the investigated ventilation shafts
are specified and the used $CH_4$ emissions inventory for comparison is introduced.

---

[1]available at http://edgar.jrc.ec.europa.eu/overview.php?v=432&SECURE=123, last access: 27.05.2020, DOI: https://data.europa.eu/doi/10.2904/JRC_
DATASET_EDGAR

[2]available at https://edgar.jrc.ec.europa.eu/overview.php?v=50_GHG, last access: 27.05.2020, DOI: https://data.europa.eu/doi/10.2904/JRC_DATASET_
EDGAR



## 2.1 CoMet measurement campaign and instrumentation

The CoMet research campaign took place in early Summer 2018 with one of its goals being the investigation of coal mining emissions from the largest European $CH_4$ emission hot spot, the USCB (between ~$18.3° - 19.2°$ E and ~$49.9° - 50.3°$ N) in Poland. $CH_4$ is emitted by over 50 coal mine ventilation shafts in that area occupying around $60 \times 40 \, \mathrm{km}^2$. However, common inventories (Crippa et al., 2020; Janssens-Maenhout et al., 2019; Scarpelli et al., 2020) provide $CH_4$ emissions only at a coarse spatial resolution of $0.1° \times 0.1°$ (translating to ~$7 \times 11 \, \mathrm{km}^2$ in the discussed area). Consequently, for optimal flight planning

and also subsequent assignment of observed $CH_4$ enhancements to specific $CH_4$ sources, the CoMet team generated a more detailed inventory. This inventory, hereafter referred to as CoMetv3 and described in further detail in Sect. 2.4, comprises annually reported $CH_4$ emissions of about $530 \, \mathrm{kt} \, CH_4 \, \mathrm{yr}^{-1}$, which are assigned to 54 exactly geolocated active ventilation shafts found in the region. Figure 2 shows the area and the ventilation shafts under consideration.

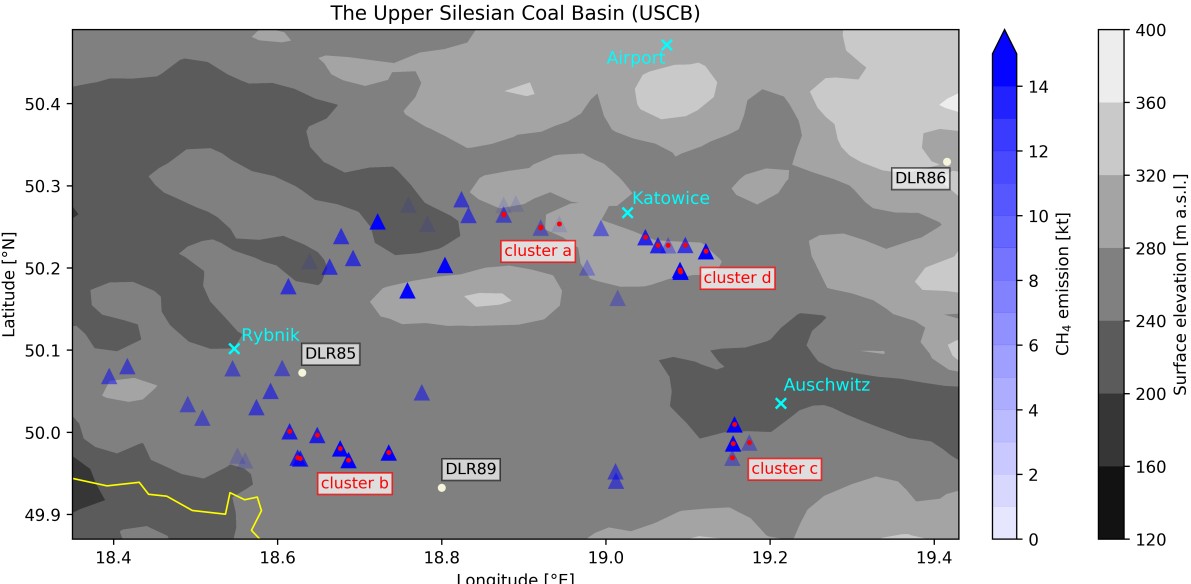

**Figure 2.** Overview of the active coal mining ventilation shafts in the Upper Silesian Coal Basin (USCB, blueish triangles). Colour intensity indicates the annual $CH_4$ emission as stated in the CoMetv3 inventory for the year 2018. Ventilation shafts filled with a red circle are investigated in this work and grouped in four clusters (see main text and Table 1 for details). Filled white circles give the locations of the three wind lidars deployed during the CoMet campaign (DLR85: $50.07025°$ N, $18.6298°$ E, at 250 m a.s.l.; DLR86: $50.3292°$ N, $19.4155°$ E, at 300 m a.s.l.; DLR89: $49.9326°$ N, $18.7998°$ E, at 270 m a.s.l.). The airport is located north of the mining region. The grey shaded area in the background indicates the terrain and the border to the Czech Republic is represented by the yellow solid line.

To investigate these $CH_4$ emissions on different scales ranging from single shafts over smaller clusters up to the entire

basin, a variety of observation platforms and instruments were deployed in the USCB during May and June in 2018. The two key instruments were the airborne passive remote sensing instrument MAMAP (operated by the University of Bremen,





Gerilowski et al., 2011) and the airborne active remote sensing instrument CHARM-F ($CO_2$ and $CH_4$ Remote Monitoring - Flugzeug, Amediek et al., 2017; Fix et al., 2015; Quatrevalet et al., 2010, operated by DLR, Deutsches Zentrum für Luft - und Raumfahrt) installed aboard a Cessna aircraft operated by the FUB (Freie Universitaet Berlin) and a Gulfstream G550 (HALO,

High Altitude and Long Range Research Aircraft) operated by DLR, respectively. Due to its long range capabilities, the HALO aircraft operated out of Oberpfaffenhofen (EDMO), Germany, whereas the FUB Cessna was deployed at the Katowice airport (EPKT), Poland, located at the northern edge of the mining area (compare to Fig. 2). Additional observations comprised airborne in situ concentrations of $CH_4$ and $CO_2$ by the FUB Cessna, by the HALO aircraft, and by a second Cessna Caravan (also operated by the DLR, Fiehn et al., 2020; Kostinek et al., 2019). The airborne observations were complemented by on

ground measurements of in situ concentrations of $CH_4$ and $CO_2$ by mobile vans (operated by AGH Krakow, IUP Heidelberg, and Utrecht University as part of the MEMO2[3] activities), stationary and mobile $CH_4$ column observations by FTS (operated by DLR, Luther et al., 2019), and wind field observations by three stationary wind lidars in that region specifically deployed for CoMet (operated by DLR, Wildmann et al., 2020). For adequate flight planning and also interpretation of the collected data sets, various model support and weather forecast systems were provided (Nickl et al., 2020, Galkowski et al., in prep).

While those results are subject to other papers either published or in preparation, the focus of the study in hand is to estimate the small scale $CH_4$ emissions of clusters of ventilation shafts by combining $CH_4$ column observations from the passive remote sensing MAMAP instrument and wind observations from the three wind lidar stations. MAMAP is a grating spectrometer, which records back scattered solar radiation from the ground while flying above the planetary boundary layer (PBL) in which the emission plumes are located. Spectra are recorded in the shortwave infrared (SWIR) region between 1590

and 1690 nm with a spectral resolution (full width at half maximum, FWHM) of around 0.9 nm. Column information of $CH_4$ is then extracted using absorption spectroscopy. The retrieved $CH_4$ column anomalies have a single-measurement precision of better than 0.4 % relative to the background column. They have, for instance, been used to estimate $CH_4$ emissions from two coal mine ventilation shafts near Ibbenbueren, Germany (Krings et al., 2013) and from landfills in Los Angeles, USA (Krautwurst et al., 2017). The precision of the instrument is therefore sufficient to investigate $CH_4$ emissions in the more

complex region of the USCB. This was also investigated by means of Observation System Simulation Experiments (OSSEs; for details, see e.g, Krautwurst et al., 2017; Gerilowski et al., 2015) before the campaign, which simulate observed $CH_4$ column anomalies based on expected emissions under various wind conditions and also considering instrumental characteristics as e.g. the measurement precision.

The wind information required for the flux estimates is derived from the three wind lidar systems (Leosphere Windcube

200S), which were deployed at three different locations in the USCB as shown in Fig. 2. They measure the vertically resolved wind field at the location of the wind lidar. Data are available as 30 minute averages in 50 m altitude bins. Additionally, the eddy dissipation rate is computed, from which the boundary layer height is estimated. The uncertainty of the wind speed is given with $0.2\,\mathrm{m\,s^{-1}}$ (Luther et al., 2019). Further details on the measurement principle and analysis are found in Luther et al. (2019), Stephan et al. (2018a, b), Smalikho and Banakh (2017), and Smalikho (2003).

---

[3]MEthane goes MObile – MEasurements and MOdelling; further details at https://h2020-memo2.eu/, last access: 21.07.2020





MAMAP observations were acquired during six flights in the USCB between 28 May and 7 June mostly before or around noon. Usual flight duration over the mining area was two to three hours each. Wind lidar observations were continuously collected throughout the entire campaign period.

## 2.2   Retrieval of column anomalies and inversion to emissions

The following sections introduce the applied algorithm to extract the desired $CH_4$ column information from the measured
MAMAP spectra, how these $CH_4$ columns are inverted to cross-sectional fluxes, and how the wind, which is needed for any flux estimate, is computed. Additionally, potential error sources of the column anomalies, the wind, and the final cross-sectional fluxes are presented.

### 2.2.1   $CH_4$ column anomalies

During a measurement flight, the MAMAP instrument typically probes the air column below the aircraft while flying above the
PBL downwind of potential emission sources. The spectra collected in this way contain the absorption features of $CH_4$ (and also $CO_2$), whose strengths depend on the amount of those gases in the atmosphere. From these features, the $CH_4$ column anomalies are retrieved using the Weighting Function Modified Differential Optical Absorption Spectroscopy (WFM-DOAS) algorithm and the $CH_4$ over $CO_2$ proxy method, which are described in detail in Krings et al. (2011) and in Sect. A1.1.

On average, the accuracy and precision of the retrieved $CH_4$ column anomalies are estimated to be around $0.10\,\%$ and
$0.22\,\%$, respectively, relative to the $CH_4$ background column for this investigated data set. The single-measurement precision is directly computed from the scatter of the measured data after applying the retrieval described in Sect. A1.1 and analyzing only observations which are not influenced by a $CH_4$ plume. The accuracy considers the influence of the terrain, such as surface elevation and surface spectral reflectance, on the retrieved $CH_4$ column anomalies, which might not be entirely accounted for during the retrieval process. A more detailed discussion of the error budget is given in Sect. A1.2.

### 2.2.2   Wind information

To describe the mass flow through a cross-section of column measurements, not only trace gas anomalies, but also wind information, are required. Ideally, the wind field is measured inside or near the emission plume simultaneously to the trace gas observations. This can be achieved for airborne in situ measurements, if the aircraft is equipped with the adequate instrumentation. In this case, trace gas and wind information are directly measured along the flight track while crossing the plume (e.g.,
Fiehn et al., 2020; Pitt et al., 2019; Ren et al., 2019; Peischl et al., 2018; Gordon et al., 2015). Since the MAMAP instrument needs to fly above the PBL, alternative sampling strategies for the wind field have been investigated in the past. This included, for instance, utilizing 3D wind fields from numerical weather prediction models (Krings et al., 2011), or splitting the measurement flight into two parts, where during the first part the trace gas column observations are collected by the remote sensing instrumentation flying above the PBL, and during the second part the wind information is collected within the PBL inside and
outside of the plume (Krautwurst et al., 2017).





For the current study, observations from the three wind lidar stations are used to estimate the prevailing winds at the location and time of the MAMAP measurement, because they are available during all six flights. As an example, Fig. 3 shows the temporal evolution of the wind speed at all three stations on 7 June. The locations of lidar stations are inlcuded in Fig. 2.

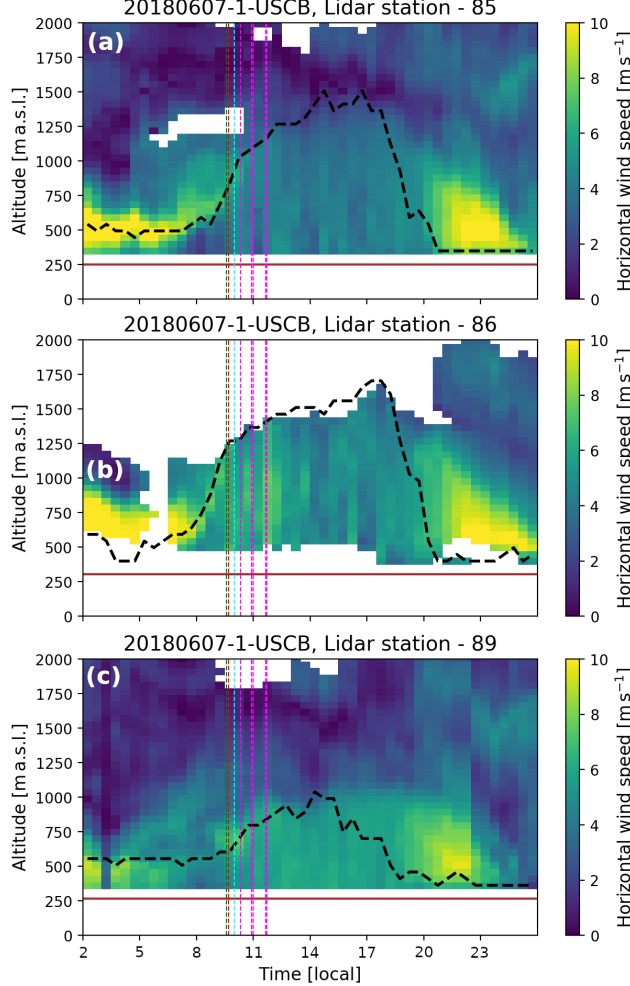

**Figure 3.** Vertically resolved wind speed measurements on 7 June from the three lidar stations deployed in the USCB. The temporal evolution of the boundary layer height is highlighted by the dashed black line. Dotted vertical lines mark the time of the different MAMAP observations/overflights on that day at the four clusters (from left to right: green: 'cluster c', red: 'cluster a', cyan: 'cluster d', magenta: 'cluster b'). Positions of the three lidars are marked in Fig. 2.

The wind speed and direction for one flight track, or one flux estimate, are computed as (time and distance) weighted-averages from all three stations, whereby only measurements within the PBL (Fig. 3, dashed black line) are considered. We assume that the plume is well-mixed within, and also confined by, the PBL. The vertical, coloured, dotted lines mark the overflight times of the single tracks on that day. For the desired mean wind speed and direction, all measurements within the



PBL are averaged for each time step and the two measurements closest in time to the overflight time are weighted according to their time difference. Finally, the values from the three stations are weighted by their distance to the flight track. A similar approach has also been chosen by Luther et al. (2019). This wind speed and direction are then used in the cross-sectional flux calculation described in the next section. As measure for the wind error, the 1-$\sigma$ standard deviation over all values as used in the average is utilized to also take into account the uncertainty caused by the variability over the basin and in time. Furthermore, this approach also covers vertical variations due to a possible wind shear or vertically unevenly distributed emissions. This leads in general to errors of ~1 m s$^{-1}$ and ~10° for wind speed and direction, respectively, which exceed the measurement uncertainty of the observations (0.2 m s$^{-1}$, Sect. 2.1) significantly. Additionally, a comparison between one of the wind lidar instruments and ultrasonic anemometers indicate biases of smaller than 0.5 m s$^{-1}$ and of around 10° for wind speed and direction, respectively (Wildmann et al., 2020). We assume that these errors are covered by our uncertainty computation because it is estimated from the standard deviation of observations from all three wind lidars, in most cases.

To also get a better overview of the large scale wind situation in the entire basin on each day, 2D wind fields are extracted from 3D WRF v3.9.1.1 reanalysis data simulations (detailed model description will be given in a separate study in the current special issue, see Galkowski et al., in prep). These fields are provided at a spatial resolution of $2 \times 2 \, \text{km}^2$ with 15 vertical levels below 3 km altitude, and high temporal resolution with instantaneous values every minute. They are used to identify unfavourable wind conditions, which would prohibit a reliable flux estimate, not obvious in the wind lidar measurements alone. To allow for a better comparison between model and wind lidars, the WRF data are averaged within the boundary layer, as calculated by the modelled PBL parametrization scheme.

Additionally, both data sets are averaged over the entire time of a measurement flight, which is on the order of two to three hours. The results are presented in Sect. 3.1.

### 2.2.3 Flux inversion

The method to derive cross-sectional fluxes has been used widely to quantify trace gas emissions not only from airborne in situ measurements (e.g., Klausner et al., 2020; Krautwurst et al., 2017; Peischl et al., 2016; Lavoie et al., 2015; Cambaliza et al., 2015; Turnbull et al., 2011; White et al., 1976) but also remote sensing column observations (e.g., Krings et al., 2018; Amediek et al., 2017; Krautwurst et al., 2017; Frankenberg et al., 2016; Krings et al., 2013) and column observations by satellite instruments (e.g., Reuter et al., 2019). The mass flow through a flight track of trace gas column observations driven by the local wind field is given by

$$F_{\text{track}} = f \cdot u \cdot cos(\alpha) \sum_i \Delta V_i \cdot \Delta x_i, \tag{1}$$

where $F_{\text{track}}$ is the resulting flux in $\text{t CH}_4 \, \text{hr}^{-1}$, $u$ is the absolute wind speed in m s$^{-1}$ as computed in Sect. 2.2.2 from the wind lidar observations, $\alpha$ is the angle between the normal of the flight track and the wind direction in degrees, $\Delta x$ is the cross-sectional length segment in m, $\Delta V$ is the retrieved $\text{CH}_4$ column anomaly in $\text{molec cm}^{-2}$ as described in Sect. 2.2.1, and $f$ is a conversion factor ($9.587 \cdot 10^{-25} \, \text{s t molec}^{-1} \, \text{hr}^{-1}$) guaranteeing the correct unit of the resulting flux $F_{\text{track}}$. The sum indicates the summation over all observations $i$ within the plume.





The dominant error sources of the computed flux $F_{\text{track}}$ arise from uncertainties or errors in the estimated wind speed ($\sim 1\,\text{m}\,\text{s}^{-1}$) and wind direction ($\sim 10°$), which can increase to up to $2\,\text{m}\,\text{s}^{-1}$ and $40°$ for specific days, the choice of the background observations, and the retrieved $CH_4$ column anomalies expressed as column anomaly precision and accuracy ($\sim 0.22\,\%$ and $\sim 0.10\,\%$, respectively, as discussed in Sect. A1.2). The error $\delta F_{\text{track}}$ of the flux $F_{\text{track}}$ of one track is computed by root sum squaring these error sources:

$$\delta F_{\text{track}} = \sqrt{\delta F_u^2 + \delta F_\alpha^2 + \delta F_{\text{bg}}^2 + \delta F_{\text{col-pr}}^2(n) + \delta F_{\text{col-ac}}^2}, \tag{2}$$

where $\delta F_u$, $\delta F_\alpha$, $\delta F_{\text{bg}}$, $\delta F_{\text{col-pr}}$, $\delta F_{\text{col-ac}}$ are the errors arising from the wind speed, from the wind direction, from the choice of the background observations, and the column anomaly precision and accuracy in $\text{t}\,CH_4\,\text{hr}^{-1}$. $\delta F_u$ and $\delta F_{\text{col-ac}}$ are computed by Gaussian error propagation of Eq. 1. $\delta F_{\text{col-pr}}(n)$ is also calculated by Gaussian error propagation taking into account its random nature by dividing the value for the estimated precision by $\sqrt{n}$, where $n$ is the number of observations within the plume. The wind direction modifies the flux via a cosine term and its error can thus not easily be calculated by error propagation. Consequently, we estimate $\delta F_\alpha$ by varying the prevailing wind direction by its estimated error on a specific day and use the difference to the 'true' flux $F_{\text{track}}$ as error estimate. The choice of the background observations is investigated by randomly selecting two-thirds of the observations from either side of the plume and computing a new background for one flight track, which is used to calculate a new flux estimate. This is done for up to 500 combinations for each side. The 1-$\sigma$ standard deviation of those fluxes is then used to estimate the error $\delta F_{\text{bg}}$.

An additional uncertainty source originates from variability in the atmospheric transport caused by turbulence and leading to varying cross-sectional fluxes if estimated from multiple overflights of the same source, which cannot be explained by source variability alone (e.g., Krautwurst et al., 2017; Matheou and Bowman, 2016, Wolff et al., in prep.). This variability, expressed as $\delta F_{\text{atm}}$, is estimated as the 1-$\sigma$ standard deviation (STD) from the overflights themselves and is then combined with the error $\delta F_{\text{tracks}}$, resulting from the errors of the single tracks, to estimate the standard error (1-$\sigma$) of the averaged flux if multiple overflights of the same source(s) are available:

$$\delta F = \sqrt{\delta F_{\text{tracks}}^2 + \delta F_{\text{atm}}^2}, \tag{3}$$

with

$$\delta F_{\text{tracks}} = \frac{\sqrt{\sum_{j=0}^{m} \delta F_{\text{track},j}^2}}{m}, \tag{4}$$

and

$$\delta F_{\text{atm}} = \frac{\text{STD}(F_{\text{track},j})}{\sqrt{m}} \tag{5}$$

where $m$ is the number of flight tracks.

## 2.3 Investigated mines and shafts

To reliably measure emissions and assign them to small clusters of coal mine ventilation shafts, MAMAP observations need to be collected in relatively close vicinity to the respective shafts. An adequate maximum distance depends, for example, on the





**Table 1.** Investigated coal mines and their annual $CH_4$ emissions based on the CoMetv3 inventory for the year 2018. The values for 2016 are only listed for comparison. The locations of clusters are shown in Fig. 2 and the position of the individual ventilation shafts are marked in the result section in Figs. 5, C1, C2, and C3.

| Cluster | Label of mine | Number of shafts [#] | $CH_4$ emission per shaft | |
|---------|---------------|----------------------|---------------------------|---|
| | | | 2018 | 2016 |
| | | | $[\,\mathrm{kt\,CH_4\,yr^{-1}}]$ | |
| a | Halemba | 3 | 3.9 | 3.3 |
| | Śląsk | 2 | 0.5 | 4.0 |
| b | Pniówek | 3 | 20.0 | 17.5 |
| | Zofiówka | 2 | 12.6 | 13.5 |
| | Borynia | 2 | 12.6 | 9.6 |
| c | Brzeszcze-a | 2 | 23.9 | 9.0 |
| | Brzeszcze-b | 2 | 4.2 | 9.0 |
| d | Wesoła | 2 | 20.5 | 16.7 |
| | Staszic | 2 | 12.9 | 9.2 |
| | Mysłowice | 1 | 16.7 | 16.7 |
| | Wieczorek-a | 1 | 10.6 | 14.7 |
| | Wieczorek-b | 1 | 5.0 | - |

complexity of the investigated area, the density of the occurring sources, and the position of the flight tracks on the different flight days. In general, the further away observations are acquired, the more complicated it is to disentangle observed fluxes from individual or groups of shafts due to potential mixing of the different plumes along their way. However, setting the focus to small clusters and primarily analyzing tracks closest to the shafts also limits the number of available observations.

Consequently, as compromise and for the purpose of this study, we only analyze flight tracks which are within ~15 km of ventilation shafts. This also reduces the probability of interference of large $CO_2$ sources, which would have, depending on position, an adverse effect on the retrieved $CH_4$ column anomalies (compare to Sect. A1.1). The drawback of this approach is that most clusters of shafts releasing $CH_4$ were only observed once during each flight. However, as observed in other studies

and as discussed in Sect. 2.2.3, fluxes estimated from multiple overflights can vary significantly as a result of turbulence, which leads to $CH_4$ column maxima and minima. To address this issue, we only try to separate and estimate emissions from clusters of ventilation shafts when at least 2 overflights are available. Additionally, the plume and background regions must be clearly distinguishable as they are selected by visual inspection. This is not the case, for example, if the flight track passes over lakes, which have a very low reflectivity in the SWIR spectral region and thus poor signal to noise ratio. Consequently observations

acquired over water bodies are thus not considered in this study.



Four clusters of ventilation shafts, illustrated in Fig. 2, were identified based on the above mentioned boundary conditions. The clusters are labelled as 'cluster a' to 'cluster d' starting in the north and counting counter-clockwise. They comprise ~40 % of all $CH_4$ mining emissions in the region according to the CoMetv3 inventory. The annual $CH_4$ emissions, the name of the mines, and the number of shafts are listed in Table 1. Depending on the position of the actual flight track, which depends on 260 the prevailing wind direction on a specific day and cloud cover and the Air Traffic Control (ATC) restrictions in that region, not all shafts of a cluster could be investigated during each flight. This led to the investigation of smaller sub-clusters, as discussed further below (Sect. 3.2).

### 2.4 CoMetv3 emission inventory

The core of the CoMetv3 inventory comprises annual $CH_4$ emissions, primarily based on data from the European Pollutant 265 Release and Transfer Register (E-PRTR) and the Polish Wyższy Urząd Górniczy (WUG, Higher Mining Administration). As in both E-PRTR and the WUG report the data is reported at the facility level, these had to be disaggregated to individual ventilation shafts for this study. Thus, we equally divided annual emissions to each shaft of the reporting mine, as more detailed data is not readily available. Such disaggregation can lead to large uncertainties, as emissions are varying due to changes in excavation activities over the year, connected to changes in mining fronts, variations in airflow driven by safety considerations 270 (including methane concentration below ground) etc. The resulting $CH_4$ emissions for 2018 are displayed for the different shafts in Fig. 2 and listed for the investigated clusters in Table 1 for the years 2016 and 2018.

However, minutely or hourly resolved emissions measured directly at the investigated shaft during the time of investigation should be optimally used for comparison to high-resolution measurements like those analyzed here. Therefore, for a subset of coal mines that agreed to provide such information, we derived hourly emissions for each shaft within the CoMet project. This 275 data is based on concentrations and airflows measured directly upstream of the outlet of the ventilation shaft. The uncertainty of these hourly emissions is estimated to be 20 % of the reported value due to lacking information about the calibration procedures and instrument precision levels. A detailed comparison between the measured hourly resolved emissions, the reported annual emissions, and the observed fluxes derived from MAMAP data gives Sect. 4.

## 3 Results

This section presents the results based on the methods and data described in Sect. 2. Initially, more and less favourable flight days are identified using PBL-averaged wind fields from WRF and the wind lidar data. Secondly, the cross-sectional fluxes for one cluster ('cluster b') are presented in detail but then summarized for the remaining clusters.

### 3.1 Wind situation over the basin

For all five measurement days (28, 29 May and 1, 6, 7 June 2018), observations from the wind lidar stations are available. 285 Figure 4 illustrates two extreme and one not so favourable case. Panel (a) shows the simulated PBL-averaged 2D wind field for 7 June between 9:30 and 11:45 local time. It exhibits a homogeneous flow from east to west with some divergence to the north

**Figure 4.** Wind situation in the USCB on three different days. Similar to Fig. 2 but complemented by the PBL averaged 2D wind field from the WRF model simulations (black arrows) and the observed wind at the three lidar stations (white arrows). Panel (a) shows favourable and (b) and (c) less favourable wind conditions on 7 June, 1 June, and 29 May, respectively.





in the northern part and to the south in the southern part of the field (black arrows). Additionally, the winds estimated from the three wind lidars (white arrows) agree well in speed as well as in direction with the prediction of the model simulation. Similar situations occur for 28 May and 6 June, which also exhibited easterly flows (see Fig. B1).

The situation changed on 29 May (c). According to the WRF simulations, the wind speed is significantly lower in some parts of the basin and more variable than on 7 June changing from an easterly flow in the middle of the basin to a south-easterly flow in the western and eastern basin. The low wind speed is also confirmed by the wind lidars observing winds of around $2\,\mathrm{m\,s^{-1}}$. Whereas winds from the western lidar (DLR85) appear to agree with the WRF simulations, those from the lidar in the east of the region (DLR86) observe significantly lower wind speeds than predicted by the model (no observations are available for the

southern lidar, DLR89, on that day). On 1 June, the wind lidars observe a strong gradient in wind speed from west to east with winds blowing from the south-south-east. This is also well captured by the WRF simulations.

    Overall, the WRF model simulations support the observations by the wind lidars. Exceptions might occur during low wind conditions.

    During low and variable wind conditions as occurring on 1 June in the south-western basin and also on 29 May, an accumu-

lation or recirculation of the emitted $CH_4$ is not entirely excluded. If clusters having a small number of shafts are investigated and observations are acquired in close vicinity to the shafts, this may be less problematic. Another limitation results from the cross-sectional flux method introduced in Sect. 2.2.3. The transport through the cross-section described by Eq. (1) must be dominated by advection and not diffusion. For wind speeds less than $2\,\mathrm{m\,s^{-1}}$, however, diffusion becomes more prominent (Sharan et al., 1996).

**3.2   Estimated cross-sectional fluxes**

The following sections present the estimated cross-sectional fluxes and their corresponding errors. 'Cluster b' was investigated during all flights and, consequently, this cluster of shafts has the most comprehensive collection of measurements. It is discussed in more detail below, followed by shorter discussions concerning the three other clusters.

**3.2.1   Cluster b**

'Cluster b' comprises 7 ventilation shafts from the three mines Pniówek (3), Zofiówka (2), and Borynia (2). They are located in the south-western part of the basin near the border of the Czech Republic. Their emissions were observed by the MAMAP remote sensing instrument during all six flights, although not all shafts were covered on all days due to the position of the flight tracks. These depended on the prevailing wind direction, cloud conditions, and ATC restrictions which also included limitations on crossing the border of the Czech Republic with the FUB Cessna.

The resulting $CH_4$ column anomalies along the different flight tracks retrieved from the observations are shown in Fig. 5. The general knowledge of the wind is shown by the wind field from the WRF model simulation. The red arrow depicts the prevailing wind derived from the lidar stations (which is also used in the cross-sectional flux method). In most cases, the derived wind directions from the lidar stations are consistent with the locations and extent of the visually observed $CH_4$ column enhancements, representing plumes, and the location of ventilation shaft(s), from which the observed $CH_4$ is most probably





emitted. Reasonable agreement between the wind lidar estimate and the position of the observed plume is even found on 29 May and 1 June, when low and variable winds prevailed. The simulated 2D wind fields match the observed plume(s) and the wind from the wind lidar stations well. The largest differences between model and observations are found on days with low wind speeds according to the wind lidar stations, namely 29 May (Fig. 4, b) and 1 June (Fig. 4, c), as already identified in Sect. 3.1. The wind speeds at 'cluster b' as derived from the lidar stations were generally around 5 to 6 m s$^{-1}$s and dropped to

around 2 m s$^{-1}$ on 29 May and 1 June.

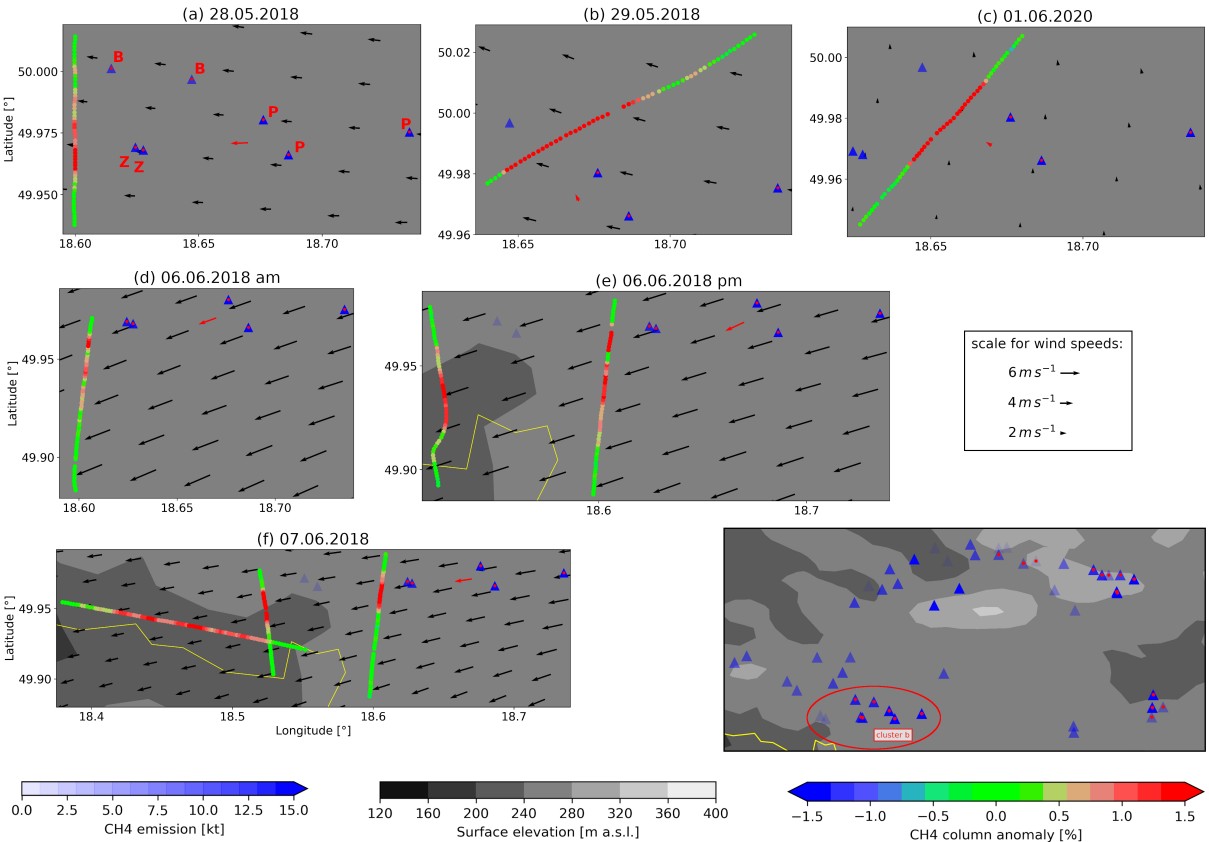

**Figure 5.** WFM-DOAS retrieval results (coloured circles) for $CH_4$ emission plume(s) originating from shaft in 'cluster b' in the southwestern part of the basin during six different overflights. For visualization only, the anomalies are smoothed by a 3-point moving average. The corresponding cross-sectional fluxes are summarized in Table 2. The grey shaded area in the background indicates the terrain and the border to the Czech Republic is represented by the yellow solid line. Black arrows illustrate the wind field based on WRF model simulations and red arrows indicate the wind at the position of the cluster/flight track at the time of the overflight as derived from the three wind lidar stations. Bluish triangles indicate reported annual emissions according to the CoMetv3 inventory and single letters are abbreviations for the ventilation shafts as listed in Table 1 (B: Borynia, Z: Zofiówka, P: Pniówek). Red dots mark the shafts responsible for the observed enhancement. On 7 June, four tracks were acquired, however, two tracks are right on top of each other. Overview map in the lower right corner is similar to Fig. 2 and highlights the investigated area and shafts by a red solid ellipse.



**Table 2.** Cross-sectional flux estimates for shaft 'cluster b' located in the south-western part of the basin during six different flights and the corresponding winds as derived from the three wind lidar stations (left part). The right part gives the errors of the five components in % of the computed flux. Footnote states which mines (number in brackets gives the number of shafts) were investigated. The stated errors of the mean flux (if more than one overflight was available) comprises the uncertainty from the error propagation of the cross-sectional flux method and the track to track variability (or atmospheric turbulence) according to Eq. (3) as discussed in Sect. 2.2.3. The last two rows give the annual [kt yr$^{-1}$] and annually scaled emissions to one hour [t hr$^{-1}$] of 2018 based on the CoMetv3 inventory (Tabel 1) for comparison with the observed averaged fluxes. For certain shafts, real hourly emissions are available and discussed in Sect. 4.

| | Flux | | | | | Errors | | | | |
|---|---|---|---|---|---|---|---|---|---|---|
| | Estimate | | | Wind | | | Wind | Background | Column | |
| | PZB$^{a)}$ | PZ$^{b)}$ | P$^{c)}$ | speed | direction | speed | direction | choice | accuracy | precision |
| | [t CH$_4$ hr$^{-1}$] | | | [m s$^{-1}$] | [°] | [%] of flux estimate | | | | |
| 28 May | 10.4 | | | 5.2 | 88 | 23 | 4 | 3 | 11 | 2 |
| 29 May | | | 8.5 | 2.2 | 151 | 27 | 42 | 4 | 7 | 2 |
| 01 June | | | 5.9 | 1.8 | 129 | 116 | 8 | 5 | 4 | 3 |
| 06 June, am | | 6.9 | | 5.6 | 70 | 18 | 9 | 7 | 17 | 4 |
| 06 June, pm | | | | | | | | | | |
|    track 1 | | 12.8 | | 6.2 | 63 | 8 | 11 | 6 | 9 | 4 |
|    track 2 | | 11.6 | | 6.5 | 68 | 11 | 6 | 9 | 12 | 6 |
| 07 June | | | | | | | | | | |
|    track 1 | | 7.7 | | 5.3 | 82 | 23 | 23 | 2 | 11 | 1 |
|    track 2 | | 8.9 | | 5.2 | 80 | 20 | 5 | 4 | 11 | 3 |
|    track 3 | | 4.7 | | 5.3 | 82 | 22 | 9 | 5 | 12 | 3 |
|    track 4 | | 11.5 | | 5.5 | 83 | 16 | 9 | 6 | 7 | 4 |
| Average | 10.4* | **9.2** | **7.0** | | | | | | | |
| Error | | | | | | | | | | |
|   [t hr$^{-1}$] | 2.7 | 1.4 | 4.1 | | | | | | | |
|   [%] | 26 | 15 | 59 | | | | | | | |
| Inventory | | | | | | | | | | |
|   [kt yr$^{-1}$] | 110 | 85 | 60 | | | | | | | |
|   [t hr$^{-1}$] | 12.6 | **9.7** | **6.8** | | | | | | | |

*based on only one single overflight.

$^{a)}$ Pniówek (3), Zofiówka (2), and Borynia (2). $^{b)}$ Pniówek (3), Zofiówka (2). $^{c)}$ Pniówek (3).





In terms of $CH_4$ emission estimates, only the flight on 28 June covered ventilation shafts from all three mines (sub-cluster 'PZB'). Only the Pniówek mine was investigated on the two days with low wind speeds (on 29 May and 1 June, sub-cluster 'P'), and Pniówek and Zofiówka together on 6 and 7 June (sub-cluster 'PZ'). The single flux estimates and their related uncertainties for 'cluster b' and its sub-clusters are summarized in Table 2 ('single' refers here to the flux of one overflight or track). The

most overflights performed on multiple days were acquired for the Pniówek and the Zofiówka shafts. The single cross-sectional fluxes originating from these two mines with five shafts vary between 4.7 and $12.8\,\mathrm{t\,CH_4\,hr^{-1}}$ with combined errors (according to Eq. (2)) of around $18\,\%$ to $34\,\%$ on the single fluxes. The error due to variability in the atmospheric transport, which needs to be considered as an additional error source for the averaged flux as discussed in Sect. 2.2.3, is at the upper end of this range with around $32\,\%$ and reduces to $12\,\%$ when accounting for the number of available flight racks (compare to Eq. (5)).

The averaged flux for this sub-cluster is $9.2\,\mathrm{t\,CH_4\,hr^{-1}}$ with a standard error of $1.4\,\mathrm{t\,CH_4\,hr^{-1}}$ (or $15\,\%$, calculated according to Eq. (3)), which compares well to the reported annual $CH_4$ emission of $9.7\,\mathrm{t\,CH_4\,hr^{-1}}$. Even for the observations under low wind conditions on 29 May and 1 June (sub-cluster 'P'), the estimated averaged flux agrees within $2\,\%$ with the annual inventory value.

As discussed in Sects. 2.2.3 and 2.3, fluxes derived from one single overflight might differ significantly from the true

emissions. The estimated flux, and also its error, on 28 May is listed for the sake of completeness and should be interpreted with caution, although it agrees within ~$20\,\%$ with the reported emissions in this case. A closer look at the inventory values and the observed averaged fluxes is given in Sect. 4.

The dominant error source (Table 2) of the single fluxes is the wind speed (and for some tracks the wind direction) followed by the accuracy of the retrieval and the choice of the background observations. The single-measurement precision of the

MAMAP instrument is mostly negligible. The error on the wind speed is usually between 0.5 and $1.2\,\mathrm{m\,s^{-1}}$, leading to errors on the estimated flux of around $10\,\%$ to $25\,\%$ assuming a wind speed of ~$5\,\mathrm{m\,s^{-1}}$. However, for example, on 1 June the magnitude of the wind was small and variable and its error is larger than the absolute value of $1.8\,\mathrm{m\,s^{-1}}$ used for the flux estimate. This leads to an error of over $100\,\%$ on the single flux estimate and explains the large standard error of over $50\,\%$ on the averaged flux for the Pniówek shafts alone (sub-cluster 'P').

### 350  3.2.2  Clusters a, c, and d

For the remaining clusters, the retrieved $CH_4$ anomalies are shown in Figs. C1, C2, and C3, and the computed cross-sectional fluxes are listed in Table C1.

Similar to 'cluster b', the derived wind directions are consistent with the position of ventilation shafts under investigation and the observed plumes. Wind speeds measured by lidar were around 5 to $6\,\mathrm{m\,s^{-1}}$. Exceptions occur on 29 June and 1 June, when

only low and variable winds were encountered having speeds of between 1.6 to $2.9\,\mathrm{m\,s^{-1}}$ according to the lidar observations.

Estimated averaged cross-sectional fluxes for 'clusters a', 'c', and 'd' range from as low as 1 to up to ~$8\,\mathrm{t\,CH_4\,hr^{-1}}$. As for 'cluster b', not all shafts of one cluster could be investigated on all days, leading to a further division into five sub-clusters in total. Standard errors on the averaged fluxes of one sub-cluster are usually around $20\,\%$. Larger errors occur during low wind conditions (e.g., at sub-cluster 'WSMW' of 'cluster d' with $46\,\%$) or if the fluxes are small and/or only a limited amount of





overflights are available (e.g., at sub-cluster 'HS' of 'cluster a' with 42 %). A more detailed comparison of the retrieved fluxes with those from inventories is given in the next section.

An example, in which the investigation of all ventilation shafts of one cluster on specific days is not feasible, is given for 'cluster c'. The flight track is located downwind of four shafts belonging to Brzeszcze towards the west on 6 and 7 June (Fig. C2). However, the plume of the northernmost shaft could not be quantitatively investigated because it was always located

directly over an area covered by lakes, which do not allow for passive remote sensing observations. During the flight on the 1 June all four shafts were covered. However, in addition to the low wind speeds, only one overflight is available and, therefore, the flux is only listed as a matter of completeness.

## 4 Comparison to inventories

As the MAMAP measurements represent a "snapshot" of the emissions of small clusters of ventilation shafts for a short time

interval, comparisons to annually resolved and/or coarsely gridded inventories should be performed carefully, and even then conclusions drawn can be of limited value. It is therefore not expected that the emissions derived from the observed cross-sectional fluxes are identical to the reported annual emissions. The reasons for fluctuations in mining emissions are diverse (compare to Sect. 2.4) and the MAMAP observations are strictly speaking only valid for the time of the overflight. Some of the measured hourly data in the CoMetv3 inventory not only indicate fluctuations from hour to hour but also differences between

the emissions from different ventilation shafts of one mine. Detailed hourly emission data were for example collected for the three Pniówek shafts for the time period between 14 May and 13 June 2018. Maximum hour to hour fluctuations reach up to ~70 % of the averaged emissions for a single shaft over the 1 month of measurements. For the entire mine, i.e. three shafts combined, fluctuations can still reach ~30 %.

Detailed hourly emissions were not only collected for the Pniówek but also for the Zofiówka shafts of 'cluster b' (see Table

3). For the observations on 29 May and 1 June, where only the Pniówek shafts were investigated and low winds prevailed, the measured averaged hourly emissions for the time of the overflights are $4.5\,\mathrm{t\,CH_4\,hr^{-1}}$ (~34 % lower than the reported annual emissions). The observed averaged flux derived from MAMAP data is $(7.0\pm4.4)\,\mathrm{t\,CH_4\,hr^{-1}}$. This flux is larger than the measured hourly emissions, however, it was recorded under low wind conditions and is only based on two overflights, both of which call for caution in its interpretation.

The measured averaged hourly emissions for the Pniówek and Zofiówka shafts, which were investigated on 6 and 7 June are $6.2\,\mathrm{t\,CH_4\,hr^{-1}}$, which is ~36 % lower than the annually reported emissions. Although reasonable winds prevailed and 7 tracks were acquired in total, the averaged observed flux based on MAMAP observations is $(9.2\pm1.4)\,\mathrm{t\,CH_4\,hr^{-1}}$ and thus, ~49 % larger than the measured hourly emissions. Additionally, the share of emissions between the three Pniówek shafts is at a ratio of about 5:2:1 on average during the time of the MAMAP observations as indicated by the measured hourly data. The mismatch

between the observed fluxes and hourly emissions might be related to missing $CH_4$ sources which are not explicitly accounted for in the hourly data. $CH_4$ is for example not only ventilated through the ventilation shafts, but also drained from excavations and transported to drainage stations in the area. Consequently, $CH_4$ is also released from the drainage system. Those emissions



**Table 3.** Comparison of observed averaged fluxes based on MAMAP data with annually reported emissions and measured averaged hourly emissions, when available. The measured averaged hourly emissions are additionally split into the contribution of the three shafts for Pniówek and two shafts for Zofiówka. See also main text for further details.

| Dates | Cluster | MAMAP | Annual | Hourly | | |
|---|---|---|---|---|---|---|
| | | | | | Pniówek | Zofiówka |
| | | | | $[\,\mathrm{t\,CH_4\,hr^{-1}}]$ | | |
| 29 May, 1 June | b, P | 7.0 | 6.8 | 4.5 | 1.8, 1.7, 1.0 | |
| 6, 7 June | b, PZ | 9.2 | 9.7 | 6.2 | 1.8, 0.9, 0.4 | 2.0, 1.2 |
| 6, 7 June | c, B | 2.9 | 3.7 | 2.8* | - | |

*value is not based on hourly data but partly composed of monthly data between 14 May and 13 June and annual data.

are included in the annually reported emissions but not in the measured hourly data. Additionally, some tracks might also be affected by the two Jastrzebie shafts which are faintly visible in Fig. 5 at around $18.57°$ E and $49.97°$ N. According to the CoMetv3 inventory, their annual emissions are reported as $0.3\,\mathrm{t\,CH_4\,hr^{-1}}$ in total and thus are negligible. However, the measured averaged hourly emissions at the time of the overflights are $\sim1\,\mathrm{t\,CH_4\,hr^{-1}}$ in total, which might influence tracks further downwind, but due to the scatter of the observed fluxes this effect cannot be investigated further. Taking into account these effects and also the standard error of the averaged observed flux derived from MAMAP data ($1.4\,\mathrm{t\,CH_4\,hr^{-1}}$) and the error of the measured hourly emissions, which is given with $\sim1.2\,\mathrm{t\,CH_4\,hr^{-1}}$ (or $20\,\%$), the two values are not significantly different.

For 'cluster c', which consists of four shafts, the CoMetv3 inventory only provides a monthly mean value for a one month period between 14 May and 13 June in 2018 for the two high emitting shafts of Brzeszcze-a but no hourly resolved data. The emissions of these shafts are given as 1.9 and $1.7\,\mathrm{t\,CH_4\,hr^{-1}}$, which are $\sim35\,\%$ lower than their reported annual value of $2 \times 2.7\,\mathrm{t\,CH_4\,hr^{-1}}$ (Table 1). For the two remaining lower emitting shafts, only the annual emissions of $2 \times 0.5\,\mathrm{t\,CH_4\,hr^{-1}}$ are available. The investigated sub-cluster 'B2' of 'cluster c' covers one Brzeszcze-a and the two Brzeszcze-b shafts, resulting in hourly emissions of $2.8\,\mathrm{t\,CH_4\,hr^{-1}}$, which agrees very well with the observed averaged flux of $(2.9\pm0.5)\,\mathrm{t\,CH_4\,hr^{-1}}$ (Table C1).

For the two remaining 'clusters a' and 'd', only the annual emissions are available. For 'cluster a', there is good agreement for the sub-cluster 'H', only observing two Halemba shafts (1.0 vs. $0.9\,\mathrm{t\,CH_4\,hr^{-1}}$, Table C1). However, for the sub-cluster 'HS', which includes two Śląsk shafts, the observed averaged flux is larger than the reported annual value by a factor of three. This might be explained by the limited amount of overflights and/or to the variability of the shaft emissions. A similar situation is found for the sub-clusters of 'cluster d'. In the case of favourable wind conditions as for sub-cluster 'WMW', annually reported emissions and observed average fluxes agree better than for less favourable conditions as for sub-cluster 'WSMW'.





## 5 Discussion

During the CoMet campaign several coal mine ventilation shafts have been investigated by means of passive remote sensing MAMAP and wind lidar observations. The focus was set to small groups of shafts to allow for a better source attribution of the measured $CH_4$ enhancements along the flight tracks and to distinguish emissions from different groups of shafts. Emissions of groups of shafts could be well separated from each other and their emissions at the time of the overpass were determined for several days.

The single cross-sectional fluxes for different clusters estimated from the different flight tracks vary from ~1 to $14\,t\,CH_4\,hr^{-1}$ for the time of the observation. Related combined errors of the single fluxes, mainly dominated by the error of the estimated wind speed and direction and the retrieved $CH_4$ columns, are between $20\,\%$ to $120\,\%$ of the respective single flux. Large errors are found, on the one hand, when the observed flux is relatively low. This implies that the emissions originate from a weak $CH_4$ source, leading to a small signal in the observed $CH_4$ column anomalies, and the error is thus dominated by the instrument's

noise or retrieval accuracy. On the other hand, large errors can also occur under low wind conditions when the error in wind speed is as large as the prevailing wind itself. However, both error contributors should not be evaluated independently because the strength of the observed $CH_4$ anomalies inversely depends on wind speed. For the current investigation, wind speeds around 4 to $6\,m\,s^{-1}$ with an estimated error of ~$1\,m\,s^{-1}$ appear to be optimal, resulting in acceptable wind errors of around $20\,\%$ on the single flux with well-detectable $CH_4$ signals in most cases.

Additional sources of error are caused by variability of atmospheric transport arising for example from turbulence. Depending on the stability of the atmosphere, observed fluxes might vary significantly from flight track to flight track even if the emission strength does not change over time. In the present study, this effect has been approximated by evaluating the standard deviation of all tracks belonging to one sub-cluster. For instance, the error which arises from our current inability to describe turbulence and other molecular mixing, which impact on plume propagation, is estimated to be $30\,\%$ of the averaged flux

(before accounting for the number of tracks) for the sub-cluster 'PZ' of 'cluster b'. This estimate is based on seven flight tracks and is therefore more reliable than the estimate for sub-cluster 'HS' of 'cluster a' based on only two flight tracks and resulting in $50\,\%$. This also means that fluxes based on only one track can significantly deviate from the true flux and should not be considered for evaluation of reported emissions. Independent of the number of tracks measured, the error arising from the lack of knowledge of the plume dispersion, which depends on turbulence and our understanding of the mixing and propagation of

the plume, is as important as our current knowledge of the wind parameters and column observations. Further research such as the use of higher resolution plume modeling is required to better understand and minimize this source of error.

The errors are significantly reduced by averaging multiple tracks. Under favourable conditions (reasonable winds, multiple flight tracks), the standard error can be reduced to below $20\,\%$ of the averaged flux. However, the standard error of the averaged fluxes can also increase to up to $60\,\%$ under less favourable conditions (low and variable winds, turbulent atmosphere, few flight

tracks, low $CH_4$ emissions).

The calculation of the cross-sectional flux (Eq. (1)) implies that a good wind estimate is as important as precise $CH_4$ column anomalies. In the presented study, winds were derived from three wind lidar stations deployed in the USCB. Although the





prevailing wind at a specific cluster was interpolated from these stations, the wind direction agrees well with the observed location of $CH_4$ enhancements. Larger discrepancies occur only on days with low and variable winds. On the one hand, this

might be attributed to missing wind observations at the southern lidar station on those days. On the other hand, a comparison to WRF v3.9.1.1 model simulations revealed that on those days the wind speed and direction have the largest variability across the basin. We infer that the number of measurements by three stationary wind lidars does not reveal the full complexity of mixing and plume propagation in these conditions. However, modelled wind fields match the wind lidar observations for the remaining days with higher wind speeds. To reflect the effect of a variable wind field across the basin also in the final result,

the error of the wind was estimated as 1-$\sigma$ standard deviation of the observed winds at the three lidar stations. This additionally captures wind shear and the lack of knowledge of the exact vertical distribution of the emissions within the boundary layer.

An important result of this study is the accurate separation of observed fluxes to specific ventilation shafts or clusters of ventilation shafts. As the MAMAP instrument observes the total atmospheric air column, measurements can also be acquired when the emission plume is not entirely vertically well-mixed within the PBL. This allows for observations closer to the

emission source than it would be possible with airborne in situ instruments. To derive reliable fluxes, they generally need to acquire concentration measurements further downwind of a source, where the emissions are well-mixed. This is at the expense of an increasing probability that plumes from different sources overlap, which complicates separation. To capture vertical inhomogeneities of emissions near the source by airborne in situ observations adequately, dense flight patterns, which are time consuming, need to be performed as, e.g., described in Conley et al. (2017). However, similar issues are also encountered for

the single nadir measurements of MAMAP when moving to larger scales due to the large number of shafts in that region. Additionally, on larger scales, emissions of unknown origin could potentially occur and complicate interpretation. To unambiguously assign measured enhancements to sources, imaging instruments observing multiple pixels across the flight in one time step and, thus, creating a 2 dimensional gapless map of the anomalies below the aircraft are needed. Examples are the AVIRIS-NG (Thorpe et al., 2017, 2016) and Mako (Tratt et al., 2014) airborne instruments, or the MAMAP 2D instrument,

which will combine MAMAP's high spectral sampling, sensitivity and specificity with imaging capability, currently being developed at the Institute of Environmental Physics (IUP), Bremen.

When evaluating MAMAP observations on the scales of clusters of shafts, one also needs to consider light path errors, which would lead to changes in the retrieved $CH_4$ column without any real change in its atmospheric concentration (compare to Sects. 2.2.1 and A1). To reduce the light path errors, the $CH_4$ over $CO_2$ proxy method was applied. This method is only

valid if the atmospheric $CO_2$ background concentration remains constant during the flight i.e. there are no significant $CO_2$ sources in the area. On small scales, $CO_2$ sources can be excluded more reliably than on larger scales. Moving to larger scales, $CO_2$ emissions, for example from power plants, could alter the desired $CH_4$ anomalies. One solution is to investigate the influence of $CO_2$ inhomogeneities by means of other types of measurements like in situ data as done in Krautwurst et al. (2017). The preferred option is, however, to use another gas with constant atmospheric concentration for normalization, such

as $O_2$ (Schneising et al., 2009; Frankenberg et al., 2006), and to become independent of a homogeneous $CO_2$ background. For that, MAMAP also measures $O_2$ in the $O_2A$ band at around $760\,\mathrm{nm}$ for normalization purposes, which will be investigated in future studies.



Since the emissions derived from the observed cross-sectional fluxes are strictly speaking only valid for the time of the overflight, and the amount of emitted $CH_4$ and the share between different ventilation shafts are variable, deviations between

observed fluxes and reported annual emissions are expected. Differences in the single cross-sectional fluxes measured on different days, which also capture the variability of the atmospheric transport, might reflect these circumstances. However, due to the large errors on single fluxes these two effects could not be fully separated. Comparison between hourly emissions and averaged observed fluxes revealed excellent agreement for 'cluster c' and good agreement for 'cluster b' considering the uncertainties and effects already discussed in Sect. 4. Comparisons to annually reported emissions of single shafts or

small clusters must be handled with caution and are hardly meaningful due to the high variability of the emissions. On larger scales, as for example investigated in Fiehn et al. (2020) who analyzed airborne in situ observations covering the entire basin, fluctuations of emissions from single shafts or even mines might cancel out.

## 6   Conclusions and summary

$CH_4$ emissions from coal mining activities are a significant contributor to anthropogenic greenhouse gas emissions and their

accurate quantification is an essential step on the way to meet the emission reductions agreed in the Paris agreement, which is part of the United Nations Framework Convention on Climate Change (UNFCCC, 2015). It addresses greenhouse gas emissions mitigation, adaptation, and finance. Consequently, an important motivation and research question for the multi-instrument and multi-platform campaign CoMet was how well $CH_4$ emissions from one of the largest coal mining areas in Europe can be quantified.

The passive airborne remote sensing instrument MAMAP acquired observations during six flights on five days between 28 May and 7 June 2018. The $CH_4$ column anomalies along the flight track were derived using the WFM-DOAS algorithm. These anomalies were combined with estimates of the wind speed and direction from three wind lidar stations, distributed in the USCB as part of the CoMet ground infrastructure, in a mass balance approach to compute cross-sectional fluxes. In total, based on the MAMAP observations, $CH_4$ emissions originating from four clusters comprising 23 ventilation shafts were studied and

successfully disentangled. Due to different positions of the flight tracks on different days, even smaller groups of shafts from each cluster could be investigated. Therefore, the four clusters were split into seven sub-clusters, excluding sub-clusters with only a single overflight, for analysis purposes.

Estimated averaged fluxes range over almost one order of magnitude from about 1 to $9\,t\,CH_4\,hr^{-1}$ with standard errors of about $15\,\%$ to $59\,\%$, whereby fluxes from single overflights of one (sub-) cluster deviated by up to $50\,\%$ from the averaged

flux. The most important error sources are the accuracy of the $CH_4$ anomaly retrieval of ~$0.10\,\%$ relative to the background column, the choice of the background area, and the error in wind speed and wind direction estimated to be ~$1\,m\,s^{-1}$ and ~$10°$, respectively. In extreme cases, when wind speed and direction were low or variable, the magnitude of the error was similar to the magnitude for the retrieved emission. However, wind speeds were usually around 5 to $6\,m\,s^{-1}$, which appears to be a favourable magnitude for estimating reliable fluxes with magnitudes larger than $1\,t\,CH_4\,hr^{-1}$. It is recommended that these

conditions are targeted during flight planning for future campaigns if remote sensing instruments with a similar sensitivity as





that of MAMAP are deployed. An additional source of error originated from atmospheric variability due to turbulence or other sources of the variation of the atmospheric air flow, preventing flux estimates from single overflights. It is reduced by averaging over multiple overflights. Targeting the same emission source more than once should therefore also be an essential part of flight planing activities.

The wind observations from the lidar stations were also compared to wind fields from WRF v3.9.1.1 model simulations to further investigate the wind situation in the USCB on flight days. Wind lidar observations and modeled wind fields agree well, except for one of the days with low wind conditions.

In the USCB region, the emissions of $CH_4$ from ventilation shafts can significantly fluctuate from day to day and even from hour to hour, as discussed in the example of single Pniówek shafts with variations of up to 70 % based on on-site
measurements. As a result, observed fluxes could substantially deviate from reported annual values. Therefore, comparison of $CH_4$ fluxes derived from different types of observations requires data acquisition at the same time. Additionally, observed fluxes should only be compared to hourly resolved data to capture the variability correctly. Where hourly data were available, they agreed with the observed fluxes. This emphasizes the need for hourly resolved inventories of anthropogenic emissions to improve top-down and bottom-up comparisons. Overall, the ventilation shafts investigated by MAMAP (excluding shafts only
investigated during a single overflight) account for around 40 % of the $CH_4$ mining emissions in the USCB when compared with the annual emissions in the CoMetv3 inventory.

Although the 1D MAMAP remote sensing instrument succeeded in estimating emissions of multiple clusters of ventilation shafts, a further breakdown into individual shafts requires a substantial increase in observations. Imaging instruments measuring multiple ground scenes simultaneously during one time step will resolve this issue in the future.

*Data availability.* The MAMAP $CH_4$ column anomalies, the observations from the Leosphere Windcube 200S wind lidar systems, the 3D WRF v3.9.1.1 reanalysis data simulations, and the CoMetv3 emission inventory are available from the authors upon request. The airborne in situ measurements aqcuired by the DLR Cessna, the FUB Cessna and the DLR HALO aircraft can be directly inquired from the authors or can be downloaded from the HALO database (https://halo-db.pa.op.dlr.de/).

*Special issue statement.* This article is part of the special issue "CoMet: a mission to improve our understanding and to better quantify the
carbon dioxide and methane cycles". It is not associated with a conference.

**Appendix A**

**A1    The WFM-DOAS retrieval**

**A1.1    Algorithm description**

For the retrieval of the desired $CH_4$ column anomalies, the WFM-DOAS algorithm (Krings et al., 2011) is applied as intro-
duced in Sect. 2.2.1. It uses simulated radiances, which are representative of the real atmosphere at the time and location of





the observation and are compared to the measured spectra. Deviations between the two, which may occur due to enhanced methane in the measurement emitted by a ventilation shaft, are then captured by scaling weighting functions. A weighting function describes the change of radiance due to a change of a selected atmospheric parameter (e.g., changing atmospheric concentrations of $CH_4$ and $CO_2$).

To simulate a reliable background model, i.e. a spectrum which is representative for the real atmosphere, and corresponding weighting functions, the model needs to be provided with several parameters that influence the simulated spectrum. In the case of the MAMAP instrument working between 1590 and 1690 nm, these are primarily vertical concentration profiles of $CH_4$, $CO_2$ and also water vapour ($H_2O$), complemented by pressure and temperature profiles. As backscattered solar radiation from the surface is measured, the spectrum is also influenced by the surface spectral reflectance and by scattering effects from

aerosols in the atmosphere. Also geometrical parameters like flight altitude, surface elevation and solar zenith angle are taken into account.

As these parameters change from flight to flight, they are adapted to the prevailing conditions and radiative transfer model (RTM) simulations are performed for each flight. Furthermore, a 2D look-up table approach is used to account for strong variations in the light path due to changing surface elevation and solar zenith angle along the flight track. The relevant input

parameters are listed in Table A1. The radiances as well as the weighting functions, which are then used as input for the WFM-DOAS retrieval, are calculated by the radiative transfer model SCIATRAN (Rozanov et al., 2014).

The retrieval yields profile scaling factors (PSFs) for the desired trace gas concentrations of $CH_4$ and $CO_2$, from which the $CH_4$ column anomalies are then computed as follows:

$$\Delta V_{CH_4} = \left( \frac{PSF_{ratio}}{\overline{PSF_{ratio}}} - 1 \right) \cdot CH_4^{abs\,col} \cdot k \tag{A1}$$

with

$$PSF_{ratio} = \frac{PSF_{CH_4}}{PSF_{CO_2}}, \tag{A2}$$

where $\Delta V_{CH_4}$ is the $CH_4$ column anomaly in $molec\,cm^{-1}$ used in the cross-sectional flux method (Eq. (1)), $k$ is a conversion factor without units derived from averaging kernels and takes into account that the sensitivity below the aircraft is around twice as high than above, $CH_4^{abs\,col}$ is the assumed background column of $CH_4$ in $molec\,cm^{-1}$, $PSF_{CH_4}$ and $PSF_{CO_2}$ are the retrieved

profile scaling factors without units, and $\overline{PSF_{ratio}}$ denotes a normalization process with observations from the local background. The formulas including the different quantities are further discussed below.

The retrieved PSFs of $CH_4$ and $CO_2$ describe the relative change in $CH_4$ and $CO_2$ in the measured spectra compared to the simulated one. If the observation was acquired over a $CH_4$ emission plume, the $PSF_{CH_4}$ is >1 and the $PSF_{CO_2}$ remains 1. However, the PSFs are not only influenced by the respective trace gas concentrations in the atmosphere but also by light

path changes resulting from, e.g., variations in flight altitude, surface elevation or enhanced scattering, not perfectly covered by the RTM simulations. These light path errors affect the absorption behaviour of both gases in a similar way due to their spectral proximity and can, therefore, be significantly reduced by applying the $CH_4$ over $CO_2$ proxy method (Krings et al.,





**Table A1.** General boundary conditions for the six flights performed during CoMet and also used for the radiative transfer model (RTM) simulations.

| Flight day | 28.05.2018 | 29.05.2018 | 01.06.2018 | 06.06.2018 | 06.06.2018 | 07.06.2018 |
|---|---|---|---|---|---|---|
| Flight time appr. (local time) | | | | | | |
| start [hh:mm] | 10:33 | 10:17 | 09:07 | 09:31 | 14:17 | 09:09 |
| end [hh:mm] | 13:02 | 12:50 | 12:03 | 12:26 | 17:12 | 11:58 |
| Solar zenith angle (SZA) [a] | | | | | | |
| min [°] | 28.0 | 28.1 | 28.8 | 27.6 | 32.8 | 28.8 |
| max [°] | 39.3 | 40.9 | 50.9 | 46.8 | 58.4 | 51.1 |
| Flight altitude [m a.s.l.] [b] | 3235 | 3205 | 3173 | 3143 | 3150 | 3160 |
| Surface elevation along flight track [c] | | | | | | |
| min [m] | 113 | 112 | 119 | 109 | 109 | 110 |
| max [m] | 436 | 427 | 427 | 471 | 475 | 464 |
| Mean column mole fractions [d] | | | | | | |
| $CH_4$ [ppb] | 1835 | 1839 | 1833 | 1836 | 1834 | 1841 |
| $CO_2$ [ppm] | 401.7 | 407.3 | 400.2 | 408.3 | 408.3 | 408.3 |
| $H_2O$ [ppm] | 4166 | 3140 | 4508 | 2517 | 2148 | 2941 |
| Aerosol scenario [e] [−] | | | urban | | | |
| Albedo [f] [−] | | | 0.18 | | | |

[a] SZA is computed from the GPS (Global Positioning System) time stamp and assigned to each observation.

[b] Flight altitude is computed as average over the entire flight from the GPS signal.

[c] Topography data is obtained from the Shuttle Radar Topography Mission (SRTM, Farr et al., 2007, version 2.1, http://dds.cr.usgs.gov/srtm/version2_1/) digital elevation model, which is assigned to each observation based on its current GPS position.

[d] The vertical atmospheric profiles are based on the U.S. standard atmosphere (USCESA, 1976), which are then adapted according to the airborne in situ observations ($CH_4$ and $CO_2$) acquired by the DLR Cessna, the FUB Cessna and the DLR HALO aircraft and the WRF-CHEM v3.9.1.1 model simulations ($H_2O$).

[e] As aerosol scenario, a standard OPAC (Optical Properties of Aerosol and Clouds, Hess et al., 1998) urban aerosol scenario is applied.

[f] The surface is assumed as a Lambertian reflector with a constant and wavelength independent surface spectral reflectance in nadir direction of 0.18, which is a common value for mid latitude vegetation and also used in previous studies (e.g, Krings et al., 2011).

2013, 2011) denoted by Eq. (A2). The drawback of this method is, however, that strong $CO_2$ sources must not be located in the measurement area and the $CO_2$ concentration remains constant during one flight, which is true on smaller scales like single shafts or small clusters of shafts, but might be invalid if the entire USCB is investigated. Finally, the PSF ratios are normalized by the local background (denoted by $\overline{\mathrm{PSF_{ratio}}}$ in Eq. (A1)) and corrected by the conversion factor $k$ to get the desired $CH_4$ column anomalies needed for the cross-sectional flux method. The local background is defined similarly to how it has been done in other publications (e.g., Krings et al., 2018; Krautwurst et al., 2017; Frankenberg et al., 2016) as observations outside of a plume in its flanks and determined by visual inspection of each single track downwind of a potential source (cluster).





**A1.2   Errors**

Errors in the retrieval of the $CH_4$ column anomalies originate from the measurement noise of the instrument or the input parameters for the RTM simulations. The measurement noise is computed as single measurement precision relative to the background column directly from the scatter of the measured data. The retrieval described above is applied and the observations, which are not influenced by a $CH_4$ plume, are used. For the currently investigated data set, this has been estimated to be 0.22 %

relative to the background column on average.

The sensitivity of the input parameters on the final $CH_4$ column anomaly is estimated by using synthetic spectra while varying the input parameters according to their typical variation during a flight as given in Table A2. As expected and already shown in earlier studies (e.g., Krings et al., 2011), the deviations in the fitted profile scaling factors easily reach some percent and, therefore, are on the same order of magnitude as those caused by actual emissions. As most of the deviations are related to

light path errors, the applied proxy method reduces these deviations significantly. Most of the remaining effects are systematic and constant along a flight track (e.g., a constant offset caused by wrongly assumed $CO_2$ or $CH_4$ background concentration, background temperature or background aerosol profiles), and are corrected by the normalization using observations outside of a plume. Parameters which may not be covered by the normalization process, but also do not fluctuate randomly along a flight track and therefore may not be entirely covered by the computed single measurement precision, are surface elevation

and surface spectral reflectance. In a worst case scenario, part of the flight track is located over an especially bright surface or over relatively high terrain (forest vs. rangeland) compared to the remaining track. In this study, the uncertainties originating from these two factors are therefore assumed to be uncorrelated and after accounting for the conversion factor k (~0.69), they potentially lead to a systematic offset of the retrieved $CH_4$ column anomaly of around 0.10 %.

In combination with the single measurement precision, they are considered in the column anomaly computation by Eq. (1).

Although the values in Table A2 are computed for the flight on 7 June, they are assumed to be valid also for the other days.





**Table A2.** Sensitivity of the retrieved profile scaling factors (PSFs) to the input parameters for the radiative transfer model (RTM) simulations according to expected variations during one flight on 7 June. The deviations for the PFSs of $CH_4$, $CO_2$ and the ratio $CH_4$ over $CO_2$ are again given relative to the background column. The parameters for the true or basic scenario are listed in Table A1, 7 June using a flight altitude of $3.16\,km$ and a solar zenith angle of $39.4°$. Not all values deviate symmetrically around $0\,\%$, therefore, the worst case scenario is always selected.

| variation in parameter | Expected deviation of PSF [%] | | |
| --- | --- | --- | --- |
| | $CH_4$ | $CO_2$ | ratio |
| Solar zenith angle [$\pm\,2°$] | $\pm 2.36$ | $\pm 2.29$ | $\pm 0.08$ |
| Surface elevation [$\pm\,30\,m$] | $\pm 0.60$ | $\pm 0.68$ | $\pm 0.08$ |
| Flight altitude [$\pm\,20\,m$] | $\pm 0.12$ | $\pm 0.12$ | $\pm 0.01$ |
| Aerosol [desert, background] | $\pm 0.16$ | $\pm 0.45$ | $\pm 0.29$ |
| Albedo [$0.05-0.50$] | $\pm 0.68$ | $\pm 0.80$ | $\pm 0.12$ |
| $H_2O$ [$\pm\,50\,\%$] | $\pm 0.03$ | $\pm 0.01$ | $\pm 0.02$ |
| $CO_2$ [$\pm\,1\,\%$] | $\pm 0.00$ | $\pm 1.00$ | $\mp 1.00$ |
| $CH_4$ [$\pm\,1\,\%$] | $\pm 1.00$ | $\pm 0.00$ | $\pm 1.00$ |
| Temperature [$\pm\,5°C$] | $\pm 1.57$ | $\pm 1.88$ | $\pm 0.31$ |



## Appendix B: Wind fields



**Figure B1.** Similar to Fig. 4, but for the remaining three flights on 28 May (a), and 6 June in the morning (b) and in the afternoon (c).





## Appendix C: Column anomalies and fluxes of clusters a, c, and d

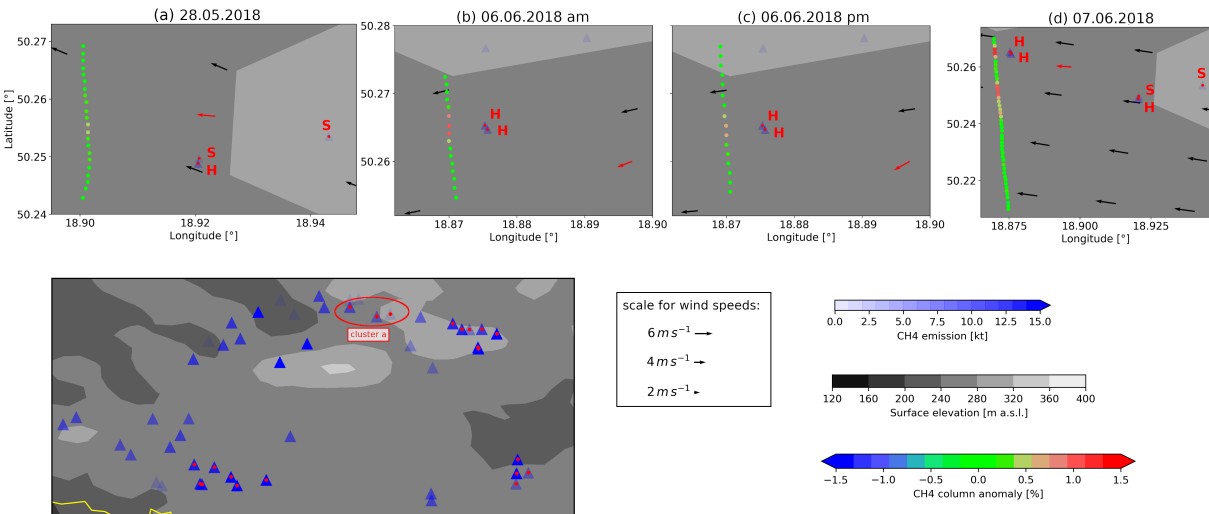

**Figure C1.** Same as Fig. 5 but for shaft 'cluster a' in the northern part of the study area (H: Halemba, S: Śląsk). The corresponding cross-sectional fluxes are summarized in Table C1.

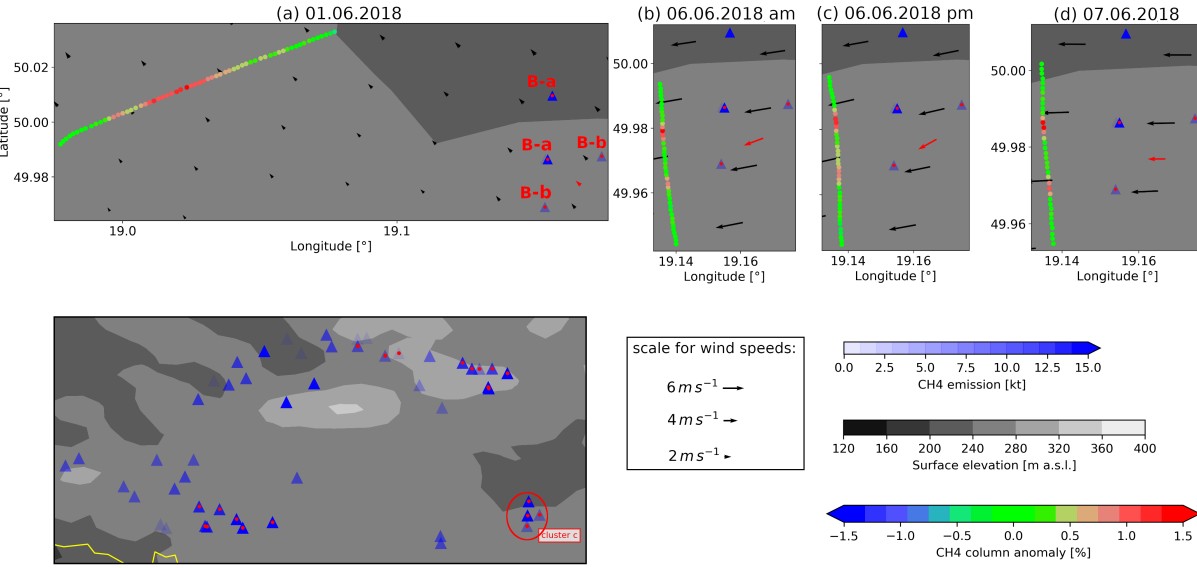

**Figure C2.** Same as Fig. 5 but for shaft 'cluster c' in the south-eastern part of the study area (B-a: Brzeszcze-a, B-b: Brzeszcze-b). The corresponding cross-sectional fluxes are summarized in Table C1.

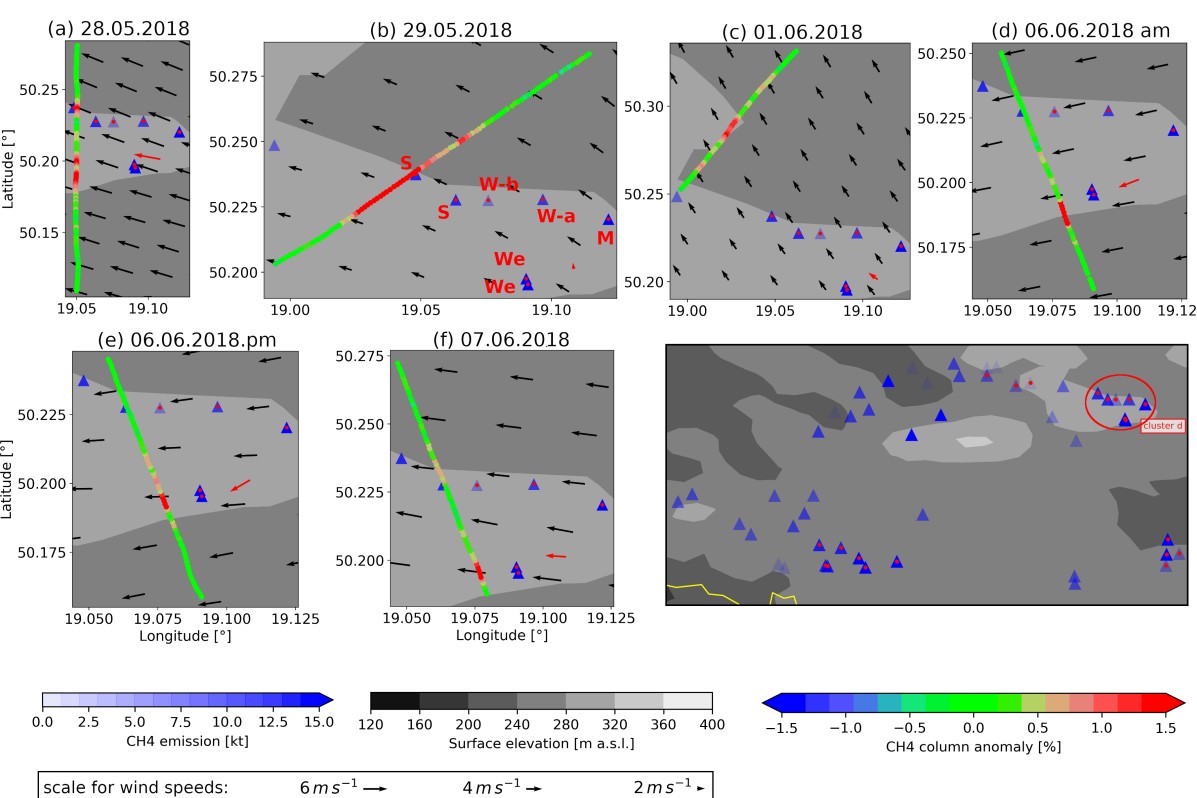

**Figure C3.** Same as Fig. 5 but for shaft 'cluster d' in the north-eastern part of the study area (S: Staszic, W-b: Wieczorek-b, W-a: Wieczorek-a, M: Mysłowice, We: Wesola). The corresponding cross-sectional fluxes are summarized in Table C1.





**Table C1.** Same as Table 2 but for 'clusters a, c, d', and without the errors of the single components.

| | Cluster a | | | | Cluster c | | | | Cluster d | | | |
| --- | --- | --- | --- | --- | --- | --- | --- | --- | --- | --- | --- | --- |
| | Flux | | Wind | | Flux | | Wind | | Flux | | Wind | |
| | HS[a] | H[b] | speed | dir. | B1[c] | B2[d] | speed | dir. | WSMW[e] | WMW[f] | speed | dir. |
| | $[\,\mathrm{t\,hr^{-1}}]$ | | $[\,\mathrm{m\,s^{-1}}]$ | $[°]$ | $[\,\mathrm{t\,hr^{-1}}]$ | | $[\,\mathrm{m\,s^{-1}}]$ | $[°]$ | $[\,\mathrm{t\,hr^{-1}}]$ | | $[\,\mathrm{m\,s^{-1}}]$ | $[°]$ |
| 28 May | 1.0 | | 6.3 | 95 | | | | | 13.7 | | 6.7 | 101 |
| 29 May | | | | | | | | | 4.9 | | 1.6 | 173 |
| 01 June | | | | | 2.8 | | 1.9 | 134 | 4.3 | | 2.9 | 126 |
| 06 June, am | | 0.9 | 5.4 | 69 | | 2.8 | 5.9 | 69 | | 7.3 | 5.5 | 69 |
| 06 June, pm | | 0.8 | 6.0 | 61 | | 3.1 | 5.9 | 61 | | 7.0 | 5.8 | 60 |
| 07 June | 2.1 | 1.1 | 5.3 | 93 | | 2.7 | 5.0 | 89 | | 5.4 | 5.2 | 94 |
| Mean | **1.6** | **1.0** | | | 2.8* | **2.9** | | | **7.6** | **6.5** | | |
| Error | | | | | | | | | | | | |
| $[\,\mathrm{t\,hr^{-1}}]$ | 0.7 | 0.2 | | | 2.9 | 0.5 | | | 3.5 | 1.2 | | |
| $[\%]$ | 42 | 22 | | | 103 | 19 | | | 46 | 19 | | |
| Inventory | | | | | | | | | | | | |
| $[\,\mathrm{kt\,yr^{-1}}]$ | 4.9 | 7.8 | | | 56 | 32 | | | 99 | 73 | | |
| $[\,\mathrm{t\,hr^{-1}}]$ | **0.6** | **0.9** | | | 6.4 | **3.7** | | | **11.3** | **8.4** | | |

*based on only one single overflight.

[a] Halemba (1), Śląsk (2).    [b] Halemba (2).

[c] Brzeszcze-a,-b (2,2).    [d] Brzeszcze-a,-b (1,2).

[e] Wesoła (2), Staszic (2), Mysłowice (1), Wieczorek-a,-b (1,1).    [f] Wesoła (2), Mysłowice (1), Wieczorek-a,-b (1,1).





*Author contributions.* SK processed the remote sensing (RS) data and analyzed the RS and wind lidar data and data from the WRF-CHEM v3.9.1.1 model simulations, computed the fluxes, and led the writing of the manuscript. KG, JB, HB and JPB contributed to the paper draft.
KG, AnF, HB, and JPB initialized the CoMet activities including the campaign in 2018. SK, KG, JB, MG, AlF, AR, TR, CG, AnF, and HB designed the daily flight plans. SK, KG, and JB collected the remote sensing and in-situ data needed for processing of the RS data. MG, AlF, AR, CG, and AnF collected in-situ data needed for processing of the RS data. NW collected and processed the wind lidar data. MG and JM performed the WRF-CHEM v3.9.1.1 model simulations. MG, JS, and JN designed the CoMetv3 inventory. All authors contributed to the interpretation of the results and the improvement of the manuscript.

*Competing interests.* The authors declare that they have no conflict of interest.

*Acknowledgements.* We gratefully acknowledge funding for the CoMet campaign by the BMBF (German Federal Ministry of Education and Research) through AIRSPACE (FK 01LK1701B and FK 390 01LK1701C), the State of Bremen, and the MPG (Max Planck Society). The work has further been supported by the German Research Foundation (Deutsche Forschungsgemeinschaft, DFG) within the DFG Priority Program (SPP 1294) Atmospheric and Earth System Research with the Research Aircraft HALO (High Altitude and LOng Range Research
Aircraft) under grant BO 1731/1-1. We also acknowledge the use of resources of Deutsches Klimarechnungszentrum (DKRZ), namely the high-performance cluster Mistral, for data storage and analysis.

We also gratefully thank Jeremy Gordon who safely piloted the FUB Cessna during the different flights and the administration of the airport Katowice, who not only provided us with a parking space for the aircraft and gave us easy access to the hanger to service our measuring instruments during the campaign, but also took care of our physical well-being before and after the flights.



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
