# Peer review of "Quantification of $CH_4$ coal mining emissions in Upper Silesia by passive airborne remote sensing observations with the MAMAP instrument during CoMet"

_Atmospheric Chemistry and Physics, 2020_

## Author Comment (AC1)

**Reviewer #3 (R#3):**

**First of all, we would like to thank the anonymous reviewer for the detailed and helpful comments. In the following, we first state the comment of the reviewer (R#3-X) and then give our response. References to the manuscript are related to the original discussion manuscript: e.g., P2L30 → page 2, line 30.**

The authors describe the methods and the results of methane emissions measurements from coal mines in the Upper Silesian Coal Basin in Poland. The paper provides a detailed description of measurement methods used and results of emissions measurement. The study is well-designed, and the paper is deserved to be published in Atmospheric Chemistry and Physics.

**Please see our detailed response as given below.**

R#3-1) The authors should better describe the key findings of the study. The article provides a lot of technical detail and, as a result, the reader cannot see the "forest" behind the "trees." The authors provide too many peculiarities, so the article could benefit from adding general conclusions about methane emissions in the basin. The article is quite long, so part of the technical material can be moved to a supplement.

**We agree with the reviewer that some parts of the manuscript might be too detailed. Therefore, we went thoroughly through each chapter trying to identify the peculiarities and shortened or removed paragraphs where appropriate. Some technical details were additionally moved to the Appendix. This applies in particular to the abstract and the Sects. 'Method and data' and 'Discussion'. For details, see the difference document.**

**In order to provide a sound comparison between measured emissions and emission inventories, we think, it is also important elaborate on details of the used technique. Some of those are instrument specific (e.g., concentration measurements by MAMAP), however others, such as deriving a reliable wind estimate, turbulence in the atmosphere and/or minimizing errors, are more widely applicable and therefore also useful for other readers.**

**As we focus on small clusters of ventilation shafts and investigated only around 40% of all shafts in that area, we cannot derive conclusions for the entire basin, e.g., that emission inventories generally over- or underestimate the coal mining emissions in that area. This was done by e.g., Fiehn et al., 2020, using airborne in-situ observations, and is mentioned in the manuscript. However, because we used remote sensing observations, we were able to fly closer to the ventilation shafts separating them and having a closer look at single or small groups of shafts (as discussed on P21L457ff). See comment R#3-3 for further details.**

**Although the entire article (including the Appendix) comprises 39 pages, we prefer to have the technical details and additional figures, which are included in the Appendix, directly in the document of the main article and not in a Supplement, which would be a separate file. The reader should have direct access to retrieval details or additional figures without needing to look it up in another file. However, should the editor or typesetters prefer to see this information published in a supplement rather than an appendix, this can be done.**

R#3-2) It would be great to have some information about coal production in the Upper Silesian Coal Basin. What is the annual coal production? What is the rank of the coal? What is the methane content of the coal? How do the mines report emissions?

**We agree and think this would indeed be some valuable background information for the interested reader. We added a short paragraph to the introductory part.**

R#3-3) P4/L6f: The key novelty of the article is the comparison of measured data with emission inventories. What can be done to improve the accuracy of emission inventories? How the result of the study can help?

**As described in R#3-1, general conclusions about the emissions over the entire basin cannot be derived. However, in this study we had the luck to have emission data with hourly resolution directly measured within the shafts – at least for a limited number of mines/shafts. They show a high variability of the emissions during the time of the campaign, which has to be considered if the accuracy of the emission inventories should be improved by independent top-down approaches. This is not only valid for the USCB but also for other regions if emissions are highly variable. In the manuscript, we therefore concluded, that "As a result, observed fluxes could substantially deviate from reported annual values. Therefore, comparison of CH₄ fluxes derived from different types of observations requires simultaneous data acquisition. Additionally, observed fluxes should only be compared to hourly data to capture the variability correctly. This emphasizes the need for hourly inventories of anthropogenic emissions to improve top-down and bottom-up comparisons." (P23L525ff)**

**Furthermore, the derived fluxes and the hourly emission data agree within their uncertainties. To further refine emissions estimates and their source positions, and to also identify interfering unknown sources, denser observation patterns are needed. This can only be achieved by imaging remote-sensing instruments with high spatial resolution (P23L532ff). This also responds to the argument that multiple overflights over one source are needed to average over the atmospheric turbulence (P23L516f).**

[revised manuscript text omitted]
 CO2 point sources with airborne lidar, Atmospheric Measurement Techniques, 14, 2717–2736, https://doi.org/10.5194/amt-14-2717-2021, 2021.

---

## Author Comment (AC2)

**Reviewer #4 (R#4):**

First of all, we would like to thank the anonymous reviewer for the detailed and helpful comments. In the following, we first state the comment of the reviewer (R#4-X) and then give our response. References to the manuscript are related to the original discussion manuscript: e.g., P2L30  $\rightarrow$  page 2, line 30.

The manuscript "Quantification of CH4 coal mining emissions in Upper Silesia by passive airborne remote sensing observations with the MAMAP instrument during CoMet" by Krautwurst et al report on coal-mine related methane emissions in Upper Silesia. The manuscript is well written and presents an analysis of a highly relevant European region in terms of methane emissions, which totals almost 1Tg/yr (if I am not mistaken). Thus, the paper is certainly worthwhile publishing. I do have a few minor comments/suggestions for improvements, as outlined below. I also want to apologize for the delay, there is no good excuse for this apart from increased absent-mindedness during the Pandemic.

**Please see our detailed response as given below.**

R#4-1) While the paper is very well written, I sometimes had the feeling that the narrative is overly long, with relatively low information density at times. This is admittedly a rather vague point and I can't put my finger on how to exactly address this but I would urge the authors to just go through every sentence and see whether some parts can't be explained more concisely or even be removed.

We agree with the reviewer and reworked/shortened some parts of the manuscript to be more concise. Among other things, the abstract and the campaign overview have been shortened, as mentioned in R#4-4 (also see comment R#3-1 from Reviewer 3). For details, see the difference document.

R#4-2) P2/L33: "atmospheric surface temperature" which is it? Also, the surface temperature is related to Climate, why not just skip it here?

**Agreed. We removed "... and the Earth's climate".**

R#4-3) P4/L67: 820kt CH4/yr, to make it easier, you could also state the equivalent total in units of ktCH4/hr in brackets.

**Agreed. We added the range also in tCH4/hr ("63 to 94 tCH4/hr"). We chose 't' and not 'kt' to be consistent with the units later used in the manuscript.**

R#4-4) P6/L109-114: Why describe all the details about things that aren't being used here? This is your specific study, not a general campaign report overview. For the sake of general readers, just focus on topics relevant to your study and shorten/cut the rest, it will make it much easier to read (at times it reads like a campaign overview report).

Agreed. We rewrote the paragraph to only focus on the instruments/observations/models used during our study.

R#4-5) P6/L123: back scattered. Suggest to change to "reflected".

Agreed. We replaced "back scattered" by "reflected".

R#4-6) P6/L126: Please provide the relevant numbers for MAMAP in terms of total FOV (and across track pixel size) as well as integration time (and this along-track pixel size).

**The relevant numbers have been added to the manuscript (P6L125).**

R#4-7) P7/L149: Well, you also measure methane above the aircraft. also, what does "typically" mean in this context here? Are there untypical conditions in which you rotate the aircraft upside-down?

The MAMAP instrument can potentially also acquire zenith sky observations [Gerilowski et al., 2011]. However, this option has not yet been investigated in detail. Here, "typical" refers to the preferred nadir observation geometry.

R#4-8) P9/L209: Speaking of units, shouldn't you have a "CH4" in there as well related to your "t"?

**Correct. We added "CH4" to the unit of the conversion factor.**

R#4-9) P11/L254: Consequently, ...

Sentence has been removed.

R#4-10) P12/L258: "The annual emissions". from CoMetv3? Be specific

**Agreed. Added "...annual emissions from the...".**

R#4-11) P15/Figure 5: I sometimes had a hard time orienting myself as to where these submaps are located on the larger maps (2++) in the beginning. I am not sure how to salvage thus but it might be easier to use identical lat/lon ranges for almost all subplots here, even if a few details are lost. at least a-e can be done that way. In addition: You plot each measurement as a circle, which all overlap in most plots. Can you actually plot them in a way that they won't overlap (or leave gaps as in b)? This would make the along-track gradients more obvious? The color-scale is also a bit unfortunate, all I can basically see is no, medium, high methane, with almost no chance of further degradation. Why have a symmetric value-range if there are almost no negative values? What are the highest values overall (most pixels are saturated). This figure is a key and powerful plot, all your main results (or the basis of them) are located here and I have the feeling that I just see necklaces that switch colors from green to red. Consider https://colorbrewer2.org/#type=sequential&scheme=OrRd&n=9 (or similar) and make sure not most values are saturated please.

We agree with the reviewer that the plot is not able to reproduce the concentration gradients along the flight tracks in detail. The plot has already gone through several iterations testing also the suggestions from the reviewer (changing lat/lon ranges or the symbols and their size). However, none of those changes addresses all points adequately and new representation problems emerge. The range of the colour scale is based on 10 years of experience with MAMAP observations and was chosen in such a way, that plumes can easily be identify in those types of plots. Therefore, it is in part intentional that one can only distinguish between no plume (greenish), a weak plume (kind of green-reddish), and a strong plume (red). Additionally, we use a symmetric scale to also represent the noise in the data in an adequate way, which scatters around 0%. However, the observations used in this study are of high quality having only little noise, whose values are entirely covered by the greenish colours of the colour bar.

However, we recognize the need for a more quantitative representation of the gradients. Thus, we added detailed plots of the CH4 column anomalies for each cross-section in Appendix C, where the reader can see the exact value for each data point (e.g., maximum enhancements). Additionally, to ease interpretation for the viewer, we added more labels to the plot.

R#4-12) P18/L376: Hourly emissions: This is very interesting! Any chance you can create time-series of those? Are there Day/night variations?

We have added a time series of the hourly emissions from the three Pniowek shafts to the manuscript. At first glance, there are no pronounced day/night variations, however, there is a strong weekly cycle for the first three weeks (till hour 500) for the total mining emissions of the investigated mine.

R#4-13) P21/L481: Is this sentence relevant for your study here? Please make your science topic the core of the paper, not other future instruments or envisioned studies. (a few of these comments are OK but given that the paper is long and the message relatively simple, it could be more concise imho).

**We agree and removed the sentence.**

**Quantification of $CH_4$ coal mining emissions in Upper Silesia by passive airborne remote sensing observations with the MAMAP instrument during CoMet**

Sven Krautwurst1, Konstantin Gerilowski1, Jakob Borchardt1, Norman Wildmann2, Michal Galkowski3,6, Justyna Swolkien4, Julia Marshall3, Alina Fiehn2, Anke Roiger2, Thomas Ruhtz5, Christoph Gerbig3, Jaroslaw Necki6, John P. Burrows1, Andreas Fix2, and Heinrich Bovensmann1

[revised manuscript text omitted]
 was supported by in situ concentration measurements of CH4 and CO2 by the FUB Cessna, by the HALO aircrafta Gulfstream G550 (HALO, High Altitude and Long Range Rese , and by a second Cessna Caravan (also operated by the DLR, ?Kostinek et al., 2019). The airborne observations were complemented by on ground measurements of in situ concentrations of and by mobile vans (operated by AGH Krakow, IUP Heidelberg, and
- 130 Utreeht University as part of the MEMO23 activities), stationary and mobile column observations by FTS (operated by DLR, Luther et al., 2 , and (also operated by the DLR, Fiehn et al., 2020; Kostinek et al., 2019). Additionally, wind field observations by three sta-

<sup>3MEthane goes MObile – MEasurements and MOdelling; further details at , last access: 21.07.2020